# Heat wave contribution to 2022's extreme glacier melt from automated real-time ice ablation readings CE1

**Aaron Cremona**[1,2], **Matthias Huss**[1,2,3], **Johannes Marian Landmann**[1,2,4], **Joël Borner**[1,5], **and Daniel Farinotti**[1,2]

[1]Laboratory of Hydraulics, Hydrology and Glaciology (VAW), ETH Zurich, Zurich, Switzerland
[2]Swiss Federal Institute for Forest, Snow and Landscape Research (WSL), Birmensdorf, Switzerland
[3]Department of Geosciences, University of Fribourg, Fribourg, Switzerland
[4]Federal Office of Meteorology and Climatology, MeteoSwiss, Zurich-Airport, Zurich, Switzerland
[5]WSL Institute for Snow and Avalanche Research SLF, Davos, Switzerland

**Correspondence:** Aaron Cremona (cremona@vaw.baug.ethz.ch)

**Abstract.** ccelerating glacier melt rates were observed during the last decades. Substantial ice loss occurs particularly during heat waves that are expected to intensify in the future. Because measuring and modelling glacier mass balance on a daily scale remains challenging, short-term mass balance variations, including extreme melt events, are poorly captured. Here, we present a novel approach based on computer-vision techniques for automatically determining daily mass balance variations at the local scale. The approach is based on the automated recognition of colour-taped ablation stakes from camera images and is tested and validated at six stations installed on three Alpine glaciers during the summers of 2019–2022. Our approach produces daily mass balance with an uncertainty of $\pm 0.81$ cm w.e. d$^{-1}$, which is about half of the accuracy obtained from visual readouts. The automatically retrieved daily mass balances at the six sites were compared to average daily mass balances over the last decade derived from seasonal in situ observations to detect and assess extreme melt events. This allows analysing the impact that the summer heat waves which occurred in 2022 had on glacier melt. Our results indicate 23 d with extreme melt, showing a strong correspondence between the heat wave periods and extreme melt events. The combination of below-average winter snowfall and a suite of summer heat waves led to unprecedented glacier mass loss. The Switzerland-wide glacier storage change during the 25 d of heat waves in 2022 is estimated as $1.27 \pm 0.10$ km$^3$ of water, corresponding to 35 % of the overall glacier mass loss during that summer. The same 25 d of heat waves caused a glacier mass loss that corresponds to 56 % of the average mass loss experienced over the entire melt season during the summers 2010–2020, demonstrating the relevance of heat waves for seasonal melt.

## 1 Introduction

The prominent and accelerating shrinkage of glaciers worldwide driven by climate change is widely documented (Church et al., 2001; Zemp et al., 2019; Hugonnet et al., 2021). The projected glacier loss will impact sea-level rise (Parkes and Marzeion, 2018; Marzeion et al., 2020), water resources (Immerzeel et al., 2020), and cryosphere-related hazards (Stoffel and Huggel, 2012). In the Alps, the substantial losses with glacier mass balances of about $-1$ m w.e. a$^{-1}$ (Huss, 2012; Davaze et al., 2020) have a significant influence on the runoff regime (Farinotti et al., 2016; Huss and Hock, 2018), thus impacting water supply (Immerzeel et al., 2010) and hydropower production (Patro et al., 2018; Schaefli et al., 2019). Glacier mass balance directly reflects fluctuations in climatic forcing (Ohmura et al., 2007), drives ice dynamics, and determines runoff generation, making it a primary indicator for glacier and climate monitoring (Zemp et al., 2015; Trewin et al., 2021). Although glacier mass balance has been studied extensively with remote sensing (Bamber and Rivera, 2007), in situ observations (Zemp et al., 2009), and modelling approaches (Hock, 2005; Hock et al., 2019), daily-scale mass balance variations remain mostly unexplored.

Glacier mass balance at the scale of entire glaciers and mountain ranges is often derived from remote-sensing products via the geodetic method (Dussaillant et al., 2018; Denzinger et al., 2021; Hugonnet et al., 2021). In it, ice volume change is obtained by differentiating two digital elevation models (DEMs) and translated to mass change by assuming a given density for the volume change (Huss, 2013). Interpreting geodetic mass balances at short timescales can be challenging, though, since the results are sensibly affected by the choice of the volume-to-mass conversion factor (Huss, 2013). Recent studies have nevertheless been using results from geodetic studies to gain insights into short-term mass balance variations (e.g. Klug et al., 2018; Vincent et al., 2021; Beraud et al., 2022; Zeller et al., 2022).

Glacier-wide seasonal and annual mass balances are routinely computed from point measurements (Ostrem and Stanley, 1969; Zemp et al., 2009, 2013; Carturan et al., 2016; Sold et al., 2016; O'Neel et al., 2019). The direct glaciological method allows for deriving accumulation and ablation components with ablation stakes, snow probes, and snow pits distributed over the glacier (Ostrem and Stanley, 1969; Dyurgerov et al., 2002; Geibel et al., 2022). Glacier-wide mass balance is then calculated by interpolating the point balances of the network over the entire surface. Methods to do so range from a simple elevation dependence to hand-drawn contours of equal mass balance to more advanced statistical methods such as kriging and distributed modelling constrained with local measurements (Dyurgerov et al., 2002; Cogley et al., 2010). Even though glacier-wide mass balance is not directly measured, it is able to capture the temporal variability in glacier mass balance (Fountain and Vecchia, 1999; Thibert et al., 2018). Several studies investigated the temporal variability in mass balance on a seasonal to monthly basis, relying on both measurements and modelling (Braithwaite, 1995; Pellicciotti et al., 2005; Huss and Bauder, 2009; Azam et al., 2014; Sold et al., 2016; Mölg et al., 2017). However, models are calibrated on observations; therefore, the temporal resolution of these approaches is often limited by the low frequency of measurements.

Recently, techniques to measure mass balance at a higher temporal resolution have emerged. Automatic weather station were employed to measure the components of the energy budget and, thus, model the glacier surface energy balance at a sub-daily resolution (Cullen et al., 2007; Fitzpatrick et al., 2017). Gugerli et al. (2019) presented a method for continuous snow-water-equivalent observations on a glacier surface during winter, relying on a cosmic ray sensor. Automated ablation stakes were developed too. A2PS contributors (2021) presented a station deployed on the glacier surface with a steel wire anchored several metres below the ice surface. As the surface melts, the station is lowered, the wire is coiled by the station, and the vertical displacement is derived. A different approach was implemented by Landmann et al. (2021), who installed automated cameras monitoring colour-coded

ablation stakes with a temporal resolution of 20 min. From the image time series, the daily melt was derived.

Because these novel methods allow measuring mass balance at a high temporal resolution, they provide new observation-based insights into short-term mass balance variations, including extreme melt events. In this study, we present an approach for the automated reading of the colour-coded ablation stakes that were presented by Landmann et al. (2021). Our approach allows deriving daily point mass balances automatically via the direct glaciological method from a time series of close-range images depicting a given ablation stake. The method is tested during the summers of 2019–2021 for six stations installed on three Swiss glaciers. The performance of the algorithm is validated against (i) visual readings of the image time series and (ii) in situ measurements. Automatically derived daily mass balances at the six sites are compared with average daily mass balances over the last decade derived from seasonal in situ observations and modelling to detect and assess extreme melt events. Extreme melt events that occurred during the summer of 2022 are investigated in more detail to evaluate the significance of heat waves for seasonal ice ablation and water runoff. Finally, we discuss the suitability of the method for the detection of extreme glacier melt events at the scale of the Swiss Alps based on point measurements.

## 2   Study sites and field data

The three test sites, i.e. Findelgletscher, Glacier de la Plaine Morte, and Rhonegletscher, are medium-sized to large Alpine glaciers located in the central and in the southwestern part of Switzerland (Fig. 1, Table 1). The three glaciers are part of the long-term monitoring programme GLAMOS (Glacier Monitoring Switzerland). The seasonal and annual mass balances are derived via the glaciological method, exploiting a network of 5 to 12 ablation stakes distributed over the glaciers' entire elevation range. In 2019, six autonomous stations equipped with a camera observing an ablation stake were installed on these glaciers in order to monitor local mass balance on a daily basis: FIN 1 (2551 m a.s.l.) and FIN 2 (3015 m a.s.l.) on Findelgletscher, PLM (2689 m a.s.l.) on Glacier de la Plaine Morte, and RHO 1 (2241 m a.s.l.), RHO 2 (2392 m a.s.l.), and RHO 3 (2589 m a.s.l.) on Rhonegletscher (Fig. 1, Table 1).

The station setup consists of a camera and an aluminium stake that is marked with tapes of different colours. The tapes have a width of 2 cm and are placed 2 cm apart (Fig. 2). The only static element of the system is the stake, which is drilled into the ice. The camera construction, instead, is placed at the glacier surface and slides along the fixed stake. The station setup thus uses the camera as a reference point. When observing the images acquired every 20 min during the ablation period, it thus appears that the stake is emerging from the ice. To automatically read the stake, all images acquired

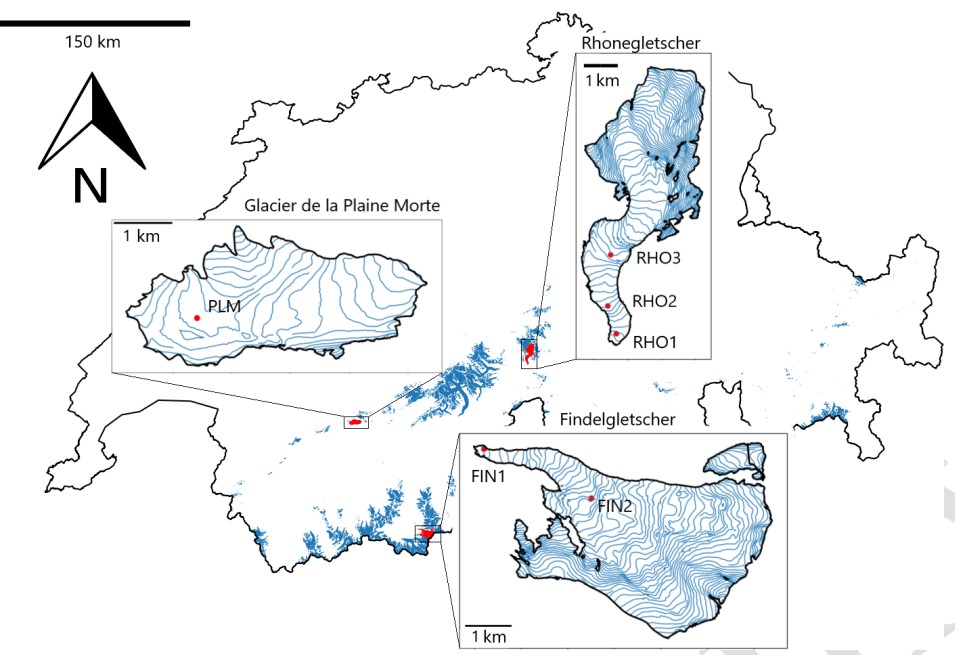

**Figure 1.** Study site overview. The locations of the six stations installed on Glacier de la Plaine Morte (station PLM), Rhonegletscher (stations RHO 1, RHO 2, and RHO 3), and Findelgletscher (stations FIN 1 and FIN 2) are indicated. Glacier outlines refer to the Swiss Glacier Inventory 2016 (Linsbauer et al., 2021), whereas region boundaries and contour lines are provided by swisstopo.

**Table 1.** Glacier and station characteristics. Area and elevation refer to the year 2019 (GLAMOS, 2021).

| | Area (km$^2$) | Elevation range (m a.s.l) | Camera stations | | |
|---|---|---|---|---|---|
| Findelgletscher | 12.7 | 2561–3937 | FIN 1 (2551 m a.s.l) | FIN 2 (3015 m a.s.l) | |
| Plaine Morte | 7.1 | 2470–2828 | PLM (2689 m a.s.l) | | |
| Rhonegletscher | 15.3 | 2223–3596 | RHO 1 (2241 m a.s.l) | RHO 2 (2392 m a.s.l) | RHO 3 (2589 m a.s.l) |

during the season are processed with a computer-vision algorithm that derives the stake emergence between pairs of subsequent images. Note that the stake emergence is the vertical movement of the stake out of the ice and is not to be confused with the concept of "emergence velocity", i.e. the difference between the local elevation change and mass balance. From the stake emergence, the mass balance is derived. The six stations were operated during the summer seasons of 2019–2022 (see Table 2 for the individual operation periods), providing point mass balances on a daily timescale.

## 3 Methods

### 3.1 Automated stake reading

We develop an algorithm that exploits computer-vision techniques to automatically derive point mass balance series from a sequence of close-range images of an ablation stake acquired during the melting season. The basis for our algorithm is provided by OpenCV, an open-source library that includes several functions designed to process images and videos (Bradski, 2000). In particular, the framework is built on matchTemplate, a function able to detect objects in an image based on a template of the object to be identified (OpenCV, 2022). The function iteratively compares the template with every portion of the target image, and the correlation between every image portion and the template is returned. A low correlation indicates that the image portion differs significantly from the template, whereas the higher the correlation, the more similar the two images are. A correlation of 1 indicates perfect agreement and can only be obtained if the template is part of the target image. The matchTemplate function may be employed in two different ways: (1) to detect a single object and (2) to detect multiple objects. In the first case, the object is detected by simply selecting the pixel with the maximum correlation value. In the second case, every pixel with a correlation above a certain threshold is depicted as a match.

For our application, we want to detect the largest number of tapes that are placed on the stake in every image. Since

**Table 2.** Operation periods of all stations during the summer seasons 2019–2021.

| Year | FIN 2 | FIN 1 | PLM | RHO 3 | RHO 2 | RHO 1 |
|---|---|---|---|---|---|---|
| 2019 | 27 Jun–17 Sep | 27 Jun–17 Sep | 19 Jun–19 Sep | 26 Jul–3 Oct | 13 Aug–3 Oct | 26 Jul–3 Oct |
| 2020 | 16 Jun–15 Sep | 16 Jun–15 Sep | 30 Jun–4 Sep | 24 Apr–24 Sep | 15 Jun–24 Sep | 15 Jun–24 Sep |
| 2021 | 8 Apr–22 Sep | 8 Apr–22 Sep | 26 Jun–18 Sep | 21 Apr–29 Sep | 21 Apr–29 Sep | 28 Jun–29 Sep |
| 2022 | not installed | 6 Apr–5 Sep | 4 Apr–23 Sep | 21 Apr–24 Sep | not installed | not installed |

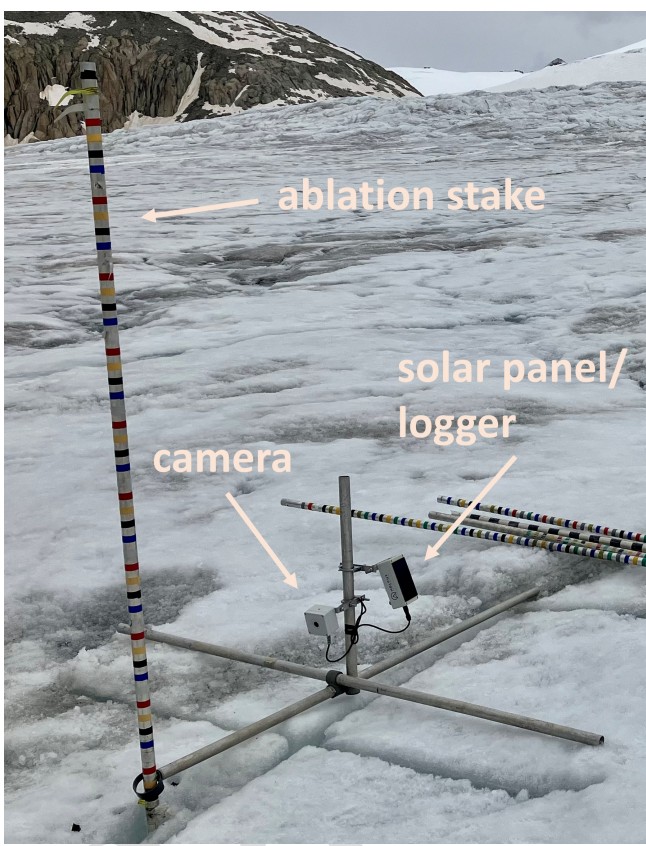

**Figure 2.** The station setup used to monitor the daily mass balance. The camera acquires pictures of the taped ablation stake every 20 min. For details of the setup, refer to Landmann et al. (2021).

a tape may look different in two subsequent images (e.g. because of changes in the illumination, caused by different cloud conditions), a relatively low correlation threshold of 0.75 is chosen. The stability of the method is also affected by the choice of template, and a tape stripe with high saturation and favourable illumination has to be preferred.

The choice of a low correlation threshold also has some drawbacks. In particular, this can cause other features in the images to be wrongly detected as tapes (Fig. 3). A two-step filtering procedure is therefore applied. The first step consists of removing duplicate matches to avoid redundancy. Such duplicate matches stem from the fact that matchTemplate operates pixel-wise and that many matches are found in the proximity of a tape. Clusters of matches are reduced to single matches according to the highest correlation within the cluster (Fig. 3). The second step consists of a collinearity check. Because all tapes are placed on the stake, i.e. on a linear feature, the correct matches are collinear. Here, we assume that the line intersecting the highest number of matches corresponds to the stake. Matches that do not intersect this line are probably located off the stake and correspond to features that are wrongly detected and thus filtered out. After this procedure, only collinear and non-redundant matches are preserved, corresponding to the tapes placed on the ablation stake.

An important limitation of matchTemplate is that it is unknown which match in a given image corresponds to which one in the subsequent image. However, most of the superimposed matches that are detected in two consecutive frames (Fig. 4d) experience a very similar displacement. This is because the tapes lie on the stake and thus move together, meaning that the displacement must be the same for every tape. Exploiting this property, the difference in position for all possible combinations is calculated and projected onto the stake axis (red line in Fig. 3) with the following equation:

$$d_k = \frac{y_{p,i-1} - y_{q,i}}{\cos \alpha_i}. \tag{1}$$

Here, $\alpha$ is the stake inclination with respect to the vertical axis, i.e. the angle between the red line in Fig. 3 and the vertical axis. The inclination is derived automatically by the algorithm. $y$ is the coordinate of a match in the current ($i$) or previous ($i-1$) image, $q \in [0, m]$ is the number of matches in image $i$, $p \in [0, n]$ is the number of matches in image $i-1$, and $d_k$ is the displacement between two matches in two consecutive images projected to the stake axis ($k \in [0, (m \cdot n)]$). Finally, the displacement occurring most frequently is taken as the stake emergence.

Figure 4 shows this procedure with an example: in Fig. 4a, 10 tapes are detected (highlighted in blue), whereas in Fig. 4b, which is the subsequent camera image, 12 tapes are detected (highlighted in yellow). By calculating the difference in the position between all combinations of tapes, the most frequent displacement, occurring eight times, corresponds to 0.668 pixels (Fig. 4d). This is taken as stake emergence during the considered time period and, thus, as vertical ice melt.

The procedure works recursively, meaning that if the stake emergence between the current image (index $i$) and the pre-

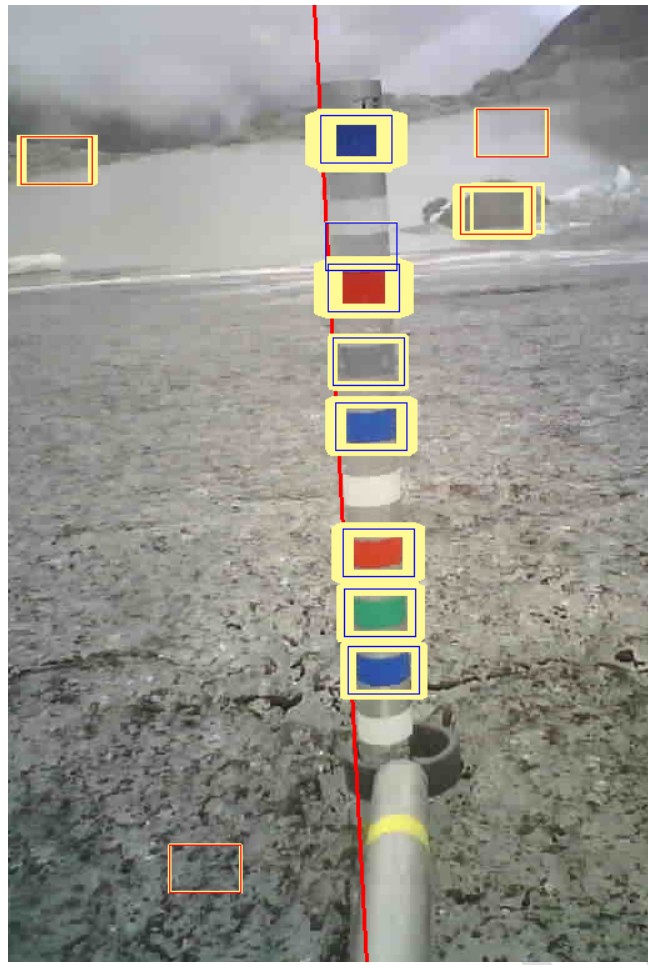

**Figure 3.** Two-step filtering procedure. (1) Clusters of matches (yellow) are reduced to single matches (red). (2) Non-collinear matches, i.e. not intersecting the red, almost vertical line, are filtered out. Tapes that are located on the stake are preserved (blue).

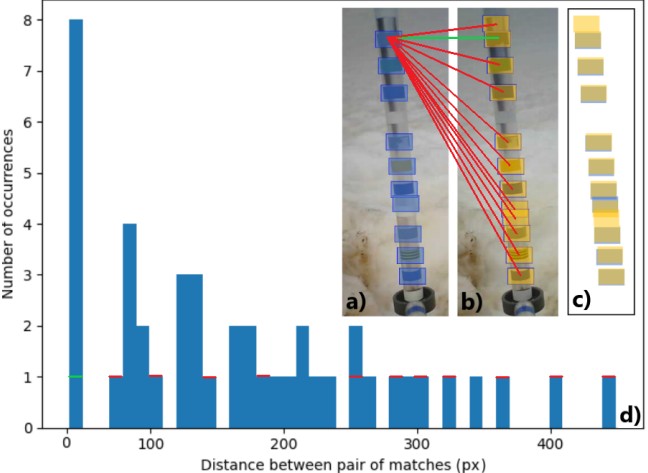

**Figure 4.** Stake emergence calculation. **(a)** Filtered matches in image $i - 1$ in blue. **(b)** Filtered matches in image $i$ in yellow. **(c)** The two sets of matches are superimposed. **(d)** Occurrence of displacements $d_k$ resulting from the difference in the positions between all possible combinations of tapes. As an example, the top match in image $i - 1$ **(a)** is compared with all the tapes in image $i$ **(b)**. The red lines show combinations of matches that provide displacements that are different from the stake emergence, whereas the green line provides the correct displacement. The displacement of each combination is reported in panel **(d)** (green and red horizontal lines). The procedure is repeated for every match in image $i$, and the most frequent displacement (0.668 pixels in the example) is taken as stake emergence.

vious image (index $i - 1$) cannot be calculated, then images $i$ and $i - 2$ are compared. This can generally be repeated up to 20 times, after which the correlation becomes too low to calculate any emergence.

Emergences in pixel units can be transformed into metric units by multiplying them with a conversion factor $c$ according to the following equation:

$$c = \frac{d_{\mathrm{ref,cm}}}{d_{\mathrm{ref,px}}}. \tag{2}$$

Here, $d_{\mathrm{ref,px}}$ is the average distance between all possible tape combinations in an image, and $d_{\mathrm{ref,cm}} = 4\,\mathrm{cm}$ is the reference distance between two tapes. In the example of Fig. 4a, the average distance between the tapes corresponds to 42.7 pixels, resulting in a conversion factor of $c = 0.094\,\mathrm{cm\,pixels^{-1}}$. This conversion method has the advantage of compensating for distortion errors. In fact, depending on the relative angle between the camera and the stake, the measured distance

between two tapes at the top of the stake can differ considerably from the distance between two tapes at the bottom of the stake. The impact of such differences is reduced by taking the average pixel distance. Converting the emergences from pixel to metric units is the last step of the algorithm to derive the stake emergences, which are then converted to mass balance through a multiplication with the density of the surface material. For bare ice, a density of $900\,\mathrm{kg\,m^{-3}}$ is assumed. For snow-covered surfaces, a density of $600\,\mathrm{kg\,m^{-3}}$ is assumed, which is derived from two snow-density measurements conducted on Glacier de la Plaine Morte in June 2019 and 2020, thus accounting for similar snow conditions, i.e. a snowpack during summer ablation (Gugerli et al., 2019). During summer snowfall, the algorithm is not able to capture accumulation because the snow which accumulates on the surface covers the camera but does not cause any relative movement between the camera and the stake. The stake emergence is thus equal to 0, which is also our ablation reading. An example of application showing the automated ablation reading on Findelgletscher can be found in the video supplement of this article.

### 3.2 Validation of automated stake reading

The automated stake reading algorithm is tested and validated with the images from the six stations acquired in the summer seasons of 2019–2021. In some cases, we observed a deterioration of the tapes over time. This causes the algorithm to struggle because of lower correlations and the correlation threshold had to be manually adjusted for the algorithm to work properly (Table 3). For some stake segments with strongly deteriorated tapes, the algorithm was unable to process the entire image time series and stopped along the way. In these cases, i.e. 15 % of the readings, the gap was filled with visual readings.

For validation, the automated readings (i.e. the outcomes of the algorithm) were compared to (i) visual readings of the same image time series (performed following Landmann et al., 2021) and (ii) in situ stake readings obtained by following the glaciological method. In situ stake readings were conducted two to five times per season in conjunction with the installation and maintenance of the stations as well as with other field campaigns. By comparing the algorithm's outcomes with the visual image readings, the daily errors in the automated approach can be computed. By comparing the algorithm's outcomes with the in situ observations, the seasonal mean absolute deviation (MAD) is derived. The results of this validation are provided in Sect. 4.1.

### 3.3 Mass balance anomalies and extreme events

The automated method provides point mass balance time series for the six stations. However, the stations are located on different glaciers and at various elevations implying that the time series cannot be directly compared against each other. In order to detect extreme melt events at a larger spatial scale, which is our target, we therefore aim at eliminating the local influences and at extracting a daily anomaly in the melt rates. This anomaly is computed with respect to an average course of the daily mass balance, where the average course refers to a longer temporal baseline. To obtain this, we rely on a combination of (i) local, seasonal mass balance measurements performed in the framework of GLAMOS at, or very close to, all six sites since at least 2010 and (ii) simple mass balance modelling for the period 2010–2020. More specifically, a daily, point-based accumulation and temperature-index melt model (Huss and Bauder, 2009; Huss et al., 2021) driven by nearby meteorological measurements is set up and constrained in every year to exactly match the observed mass balance on the date of both the April and the September survey. This means that the seasonal mass balance variations are directly provided by the direct measurements, while the daily variations in between the surveys are given by the mass balance model.

Although the stations' locations are very close to the GLAMOS stakes (for which the long-term average mass balance is available), potential biases need to be corrected for.

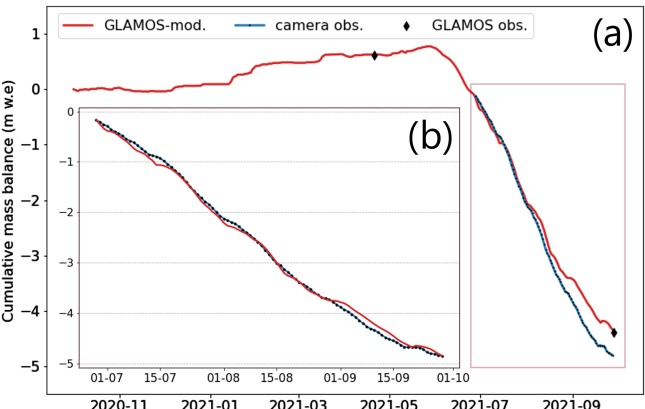

**Figure 5.** Bias correction for matching the long-term average daily mass balance to the automatically extracted daily ablation. **(a)** Cumulative long-term average daily mass balance based on seasonal observations (black diamonds) combined with modelling (red) and observed daily mass balance at the station (blue). The example refers to RHO 1 during the period 1 October 2020–1 October 2021. Note that the observed daily mass balance is only available during the summer season. **(b)** Bias-corrected modelled and observed daily mass balances during the period 29 June–29 September 2021.

To correct for these potential biases, the average daily mass balance derived from the GLAMOS stakes during the summer period is adjusted to match the result derived for the automated reading of the camera observations during the respective years (Fig. 5). For example: for the hydrological year 2021 and station RHO 1, summer melt rates based on automated readings are slightly more negative than the ones inferred from combining the seasonal observations at the nearby reference stake and the modelling (Fig. 5a). To correct for this bias, the daily difference is computed and superimposed on the long-term average daily mass balance curve as follows (see Fig. 5b for illustration):

$$\dot{b}_{i,j,\text{GLAMOS,corr}} = \dot{b}_{i,j,\text{GLAMOS}} + (\overline{\dot{b}_{j,\text{cam}}} - \overline{\dot{b}_{j,\text{GLAMOS}}}). \quad (3)$$

Here, $\dot{b}_{i,j,\text{GLAMOS}}$ is the melt rate at day $i$ of year $j$ that is derived from GLAMOS' seasonal measurements, $\overline{\dot{b}_{j,\text{cam}}}$ is the average daily melt rate over the summer of year $j$ that is derived from the camera observations, and $\overline{\dot{b}_{j,\text{GLAMOS}}}$ is the average of $\dot{b}_{i,j,\text{GLAMOS}}$ over the summer of year $j$.

For each of the 4 years with camera observations, i.e. 2019–2022, we calculate the average daily difference between the melt rates derived from camera observations and the melt rates derived from seasonal measurements of GLAMOS (Eq. 4):

$$\Delta\dot{b}_j = \overline{\dot{b}_{j,\text{cam}}} - \overline{\dot{b}_{j,\text{GLAMOS}}}. \quad (4)$$

This yields four bias values $\Delta\dot{b}_j$, i.e. one value per year, that are averaged and superimposed on the long-term average daily mass balance (Eq. 5):

$$\overline{\dot{b}_{i,2010–2020,\text{GLAMOS,corr}}} = \overline{\dot{b}_{i,2010–2020,\text{GLAMOS}}} + \overline{\Delta\dot{b}_j}. \quad (5)$$

**Table 3.** Correlation threshold used to process the image time series during the seasons 2019–2022. Stations FIN 2, RHO 2, and RHO 1 were not operated during the summer 2022 and thresholds are therefore not given. For PLM in 2022, the strong degradation of the tapes only enabled visual readings (as opposed to automated ones).

|      | FIN 2 | FIN 1 | PLM  | RHO 3 | RHO 2 | RHO 1 |
| ---- | ----- | ----- | ---- | ----- | ----- | ----- |
| 2019 | 0.75  | 0.75  | 0.75 | 0.75  | 0.75  | 0.75  |
| 2020 | 0.70  | 0.75  | 0.67 | 0.70  | 0.75  | 0.75  |
| 2021 | 0.75  | 0.70  | 0.70 | 0.70  | 0.70  | 0.70  |
| 2022 | –     | 0.69  | –    | 0.72  | –     | –     |

This procedure ensures the correction of the biases resulting from location differences between the stakes of GLAMOS and those used for the automated mass balance reading.

To compare the mass balance signals across the different stations, the daily mass balance anomaly $\Delta\dot{b}_{i,j}$ is calculated for each station (Eq. 6):

$$\Delta\dot{b}_{i,j} = \dot{b}_{i,j,\mathrm{cam}} - \overline{\dot{b}_{i,2010\text{–}2020,\mathrm{GLAMOS,corr}}}. \tag{6}$$

The mass balance anomaly is directly comparable across stations and can be used to define an extreme melt event, i.e. an event in which the anomaly exceeds a certain threshold for all stations. We set this threshold to the 85 % quantile of the negative anomalies observed during the years 2019–2021. The year 2022 was omitted when defining the threshold because of the extreme melt rates that do not represent normal conditions. Although in statistical analysis the 97.5 % or the 99.5 % quantiles are often chosen to define extreme events (Friederichs, 2010; Friederichs et al., 2018), we chose the 85 % quantile because we aim at detecting prolonged periods with high melt rates rather than only a few isolated days with maximal melt. The 85 % quantile corresponds to $-2.55\,\mathrm{cm\,w.e.\,d^{-1}}$, and the event is classified as extreme when the mean anomaly across the stations exceeds this threshold. Because the stations have different operation periods (Table 2), the mean anomaly may be computed from a different number of stations and consequently be affected. Note that the mean mass balance anomaly at the point scale does not directly allow the investigation of the influence of the heat waves on glacier melt at the regional scale, and upscaling is therefore needed.

## 3.4 Upscaling to regional mass balance

To derive the regional, i.e. Switzerland-wide, daily glacier mass balance over summer of 2022, we combine the information at the point scale, i.e. the mass balance anomalies derived from the different stations, with information about the regional mass balance. We rely on data sets of glacier-wide seasonal mass balance for 20 glaciers (GLAMOS, 2021), a daily distributed mass balance model for these glaciers (Huss et al., 2021), and multi-decadal ice volume change (Fischer et al., 2015) as well as repeated inventories of surfaces areas for all glaciers (Linsbauer et al., 2021). First,

glacier-specific anomalies in glacier-wide mass balance relative to the reference period 2010–2020 are extrapolated to all unmeasured glaciers based on inverse-distance weighting, including a higher weight for observations on the same side of the main Alpine weather divide (north/south). These anomalies are then superimposed on long-term average mass change rates available for every Swiss glacier from Fischer et al. (2015). This yields a series of annual glacier volume and mass changes at the scale of the Swiss Alps (see also Grab et al., 2021). Furthermore, daily time series of glacier-wide mass balance are available for the 20 glaciers with detailed measurements based on a combination of seasonal observations with a distributed mass balance model (Huss et al., 2021; GLAMOS, 2021). We attribute the modelled daily mass balance of the closest of the 20 glaciers with direct mass balance measurements to every glacier of the most recent Swiss glacier inventory (Linsbauer et al., 2021). Subsequently, we match the daily time series to the glacier-specific annual mass balance as originating from the procedure described above (Grab et al., 2021). This is achieved by equally attributing the misfit between the cumulative daily series and the glacier-specific annual mass balance to all days of the summer season (June–August). Overall, this procedure provides the average daily course of glacier mass balance as a mean over the period 2010–2020 for every single glacier.

To calculate the regional-scale mass balance for the summer of 2022, we combine the average regional mass balance 2010–2020 with the mean point mass balance anomaly calculated from the three stations operative in 2022 (Eq. 7).

$$\dot{B}_{i,2022} = \overline{\dot{B}_{i,2010\text{–}2020}} + \Delta\dot{b}_{i,2022}, \tag{7}$$

where $i$ is day of the year, $\dot{B}_{i,2022}$ is the computed regional mass balance for day $i$ of 2022, $\overline{\dot{B}_{i,2010\text{–}2020}}$ is the average regional mass balance of period 2010–2020 for day $i$, and $\Delta\dot{b}_{i,2022}$ is the mean point mass balance anomaly for day $i$ derived from the three stations providing daily measurements. This procedure allows the Switzerland-wide glacier mass balance over the summer of 2022 to be estimated on a daily scale. Finally, we compute the Switzerland-wide glacier storage change, which accounts for water that is produced from snowmelt and ice melt on glacier surfaces over the melt season. The glacier storage change is computed by multiplying the Switzerland-wide glacier mass balance with

the glacierized area in Switzerland according to the last inventory (2016; Linsbauer et al., 2021).

### 3.5 Definition of heat waves

To investigate the effect of heat waves on glacier melt at the scale of the Swiss Alps, we follow the definition of Kyselỳ (2002). According to Kyselỳ (2002), heat waves are defined as consecutive periods of at least 3 d during which the average daily maximum temperature is higher than 30 °C and the daily maximum temperature on individual days does not fall below 25 °C. This second part of the definition allows for hot periods separated by short, minor drops in temperature to be classified as one heat wave. Note that at the meteorological stations that we considered, the maximum daily temperature during the heat waves never dipped below 30 °C; therefore, the second part of the definition was not used. The definition of Kyselỳ (2002) considered the temperature at a single station. However, because we are interested in having a representative result at the Switzerland-wide scale, we considered the average temperature measured at different sites for defining heat waves. These four sites were the meteorological stations in Lugano (273 m a.s.l.), Chur (556 m a.s.l.), Zurich (426 m a.s.l.), and Sion (482 m a.s.l.) because they are scattered over Switzerland and cover a reasonable altitudinal range (in which we can expect temperatures above 30 °C). Hence, we assume the average temperature across the four stations to be representative of the regional scale.

## 4 Results

### 4.1 Performance of the automated algorithm

The cumulative mass balance obtained from the automated readings is very consistent with the visual readings and the in situ observations (see example of station RHO 1 in Fig. 6a–c). The good performance of the algorithm is proven by the boxplots showing the distribution of the daily deviations from the visual readings (Fig. 6d–f). In 2019, the median daily deviation is 0.06 cm and the mean deviation is −0.01 cm (Fig. 6d), in 2020, the values are −0.04 and −0.11 cm, respectively, and in 2021, they are −0.33 and −0.35 cm, respectively (Fig. 6f). The larger deviations in 2021 are likely a consequence of the tape degradation over time, which causes the tapes to loose colour and to partially detach from the stake. Of note is the fact that the daily deviations are mostly negative, which might suggest that the automated algorithm overestimates melt when compared to the visual readings. However, the comparison of the automatically retrieved cumulative mass balances with the reference mass balance (e.g. visual readings and in situ observations) for all the six sites shows that the algorithm does not overestimate melt systematically (Fig. 7). With a mean absolute deviation for the cumulative seasonal melt of 16 cm, Fig. 7 shows a good correspondence of the automatically retrieved

mass balances with both the visual readings and in situ observations.

### 4.2 Mass balance anomalies and extreme melt events

To detect and assess extreme melt events, the daily observations obtained with the automated method were compared with the long-term average mass balance, yielding daily mass balance anomalies (Fig. 8). According to our definition (see Sect. 3.3), an extreme melt event occurs when the anomaly averaged over all the stations exceeds −2.55 cm w.e. d$^{-1}$.

In 2019, 11 extreme events were identified (Fig. 8a). They are distributed more or less evenly over the summer and coincide with high air temperatures, as indicated by meteorological records from MeteoSwiss (Meteoswiss, 2020). In 2020, 14 extreme melt events occurred (Fig. 8b). Similar to the year 2019, the high-melt days are distributed over the entire summer, although in 2020 more events (ca. 50 %) are found in August. In 2021, only three extreme melt events occurred (Fig. 8c). The year was favourable for the glaciers, with only few days with extreme melting in mid-August and with overall melt close to the decadal average. This is very much in contrast to 2022, when 23 events of extreme melt occurred (Fig. 8d). During most of the summer, the anomalies are very negative, exceeding −1.5 cm w.e. d$^{-1}$ for a total of 54 d. In addition, various prolonged periods of high melt occurred throughout the summer. The comparison of the 4 years evidences that the summer of 2022 saw almost as many extreme melt events as the 3 previous years combined.

### 4.3 Impact of summer 2022 on Switzerland-wide glacier storage change

To further investigate the influence of extreme melt events and the significance of heat waves on the glacier melt in summer 2022, the daily observations were spatially upscaled to estimate the total storage change in all Swiss glaciers (see Sect. 3.4 for methods). According to our definition of heat waves (see Sect. 3.5), three heat waves occurred during the summer of 2022 for a total of 25 d. The first wave occurred between 17–21 June, the second between 14–26 July, and the third between 31 July and 6 August.

Except for August, the Switzerland-wide glacier storage change for summer 2022 was significantly higher than the average of the past decade (Fig. 9). Glacier melt was particularly high during the heat waves in June and July, during which 13 out of 23 extreme melt events occurred. During the August heat wave, instead, melting was not particularly extreme. This is because August melt rates were high during the past decade too, meaning that the melt rate anomalies for August 2022 do not exceed the threshold we use to define extreme melt events.

In addition to the above, Fig. 9 shows 10 extreme melt events that occur outside of the heat wave periods. These events are at the beginning and at the end of the season, i.e.

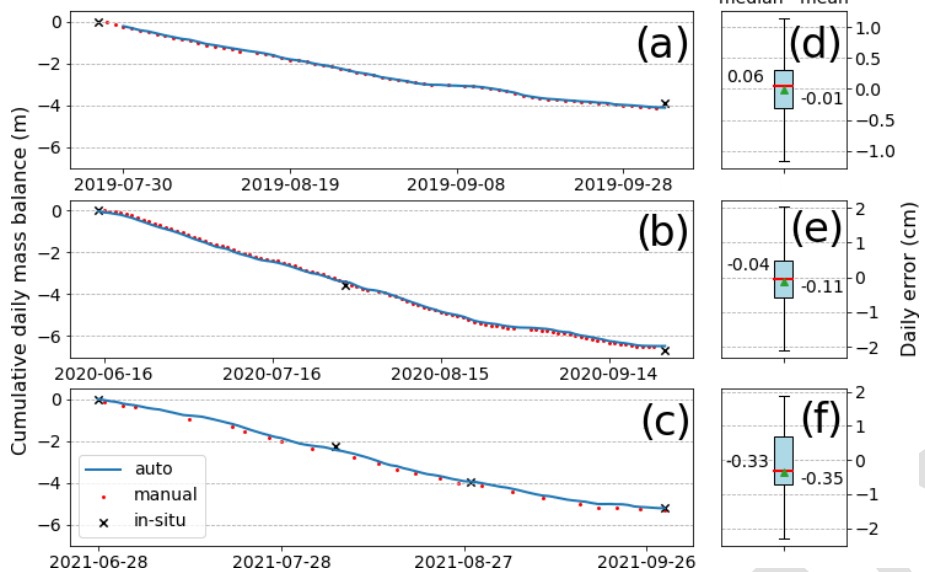

**Figure 6.** Validation of the automated mass balance readings for the station RHO 1. The cumulative daily mass balance for the years 2019 **(a)**, 2020 **(b)**, and 2021 **(c)** is shown. The mass balance from the automated method (blue) is validated against visual readings of the images (red) and in situ observations (black). Panels **(d)**–**(f)** show daily deviations with respect to the visual readings over the seasons 2019, 2020, and 2021, respectively.

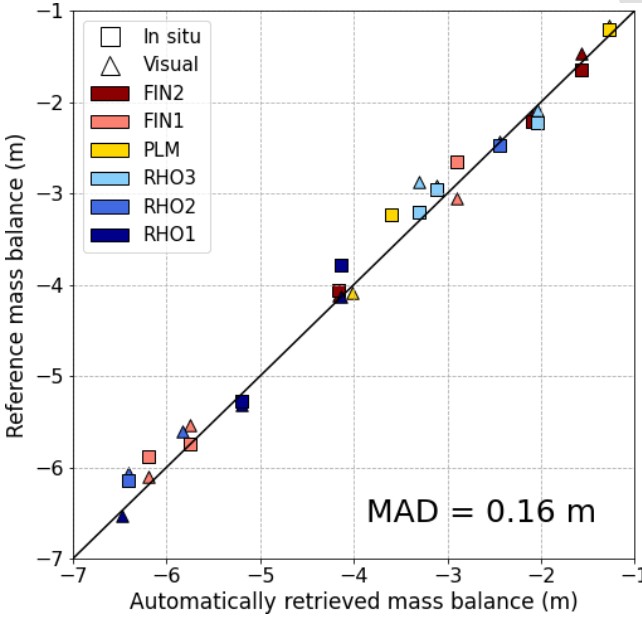

**Figure 7.** Evaluation of the automatically retrieved cumulative mass balances against the reference mass balance readings, i.e. visual readings and in situ observations. Results refer to the six stations installed over the summer seasons of 2019–2021. The seasonal mean absolute deviation (MAD) is 0.16 m.

in June and September, when maximal daily air temperatures remain below 30 °C most of the time and thus do not lead to a heat wave according to our definition. The temperatures recorded during June and September 2022 were significantly higher than in June–September of the past decade and thus caused extreme melt for several days. However, before the first heat wave (which started on 17 June 2022) and during September 2022, the temperatures were never high enough to be categorized as heat waves.

For the period from 10 June to 15 September 2022, we estimate a total storage change for all Swiss glaciers of $3.63 \pm 0.26\,\mathrm{km}^3$ w.e. (see Sect. 5.2 for the uncertainty analysis). This is about 60 % more than the average storage change in the past decade during the equivalent period ($2.25 \pm 0.10\,\mathrm{km}^3$ w.e.). The annual glacier storage change in 2022 was about 3 times higher than usual, due to exceptionally low winter accumulation in addition to the high summer ablation (GLAMOS, 2022). During the 25 d of heat waves that occurred in summer 2022, glacier storage change provided $1.27 \pm 0.10\,\mathrm{km}^3$ of water. This corresponds to 35 % of the total glacier storage change in summer 2022 and to 56 % of the average summer storage change over the past decade. This demonstrates the significance of extreme melt events occurring during heat waves regarding seasonal glacier melt.

## 5 Discussion

### 5.1 Spatial correlation of point observations

The approach we use for upscaling the mass balance at the regional scale is simple, and we thus investigate the suitability of using point measurements to detect extreme glacier melt events at the regional scale. To do so, we consider the daily

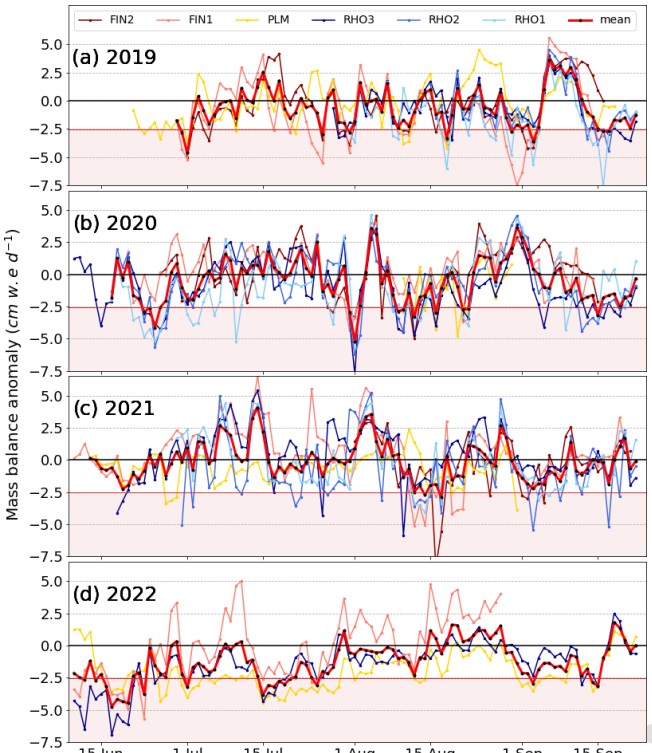

**Figure 8.** Daily mass balance anomalies for the stations on Findelgletscher (reddish CE2 colour), Glacier de la Plaine Morte (yellow), and Rhonegletscher (blueish colour) during the summer seasons of **(a)** 2019, **(b)** 2020, **(c)** 2021, and **(d)** 2022. The mean anomaly for all stations is shown in red (thick line). With our definition (Sect. 3.3), extreme events occur when the mean anomaly exceeds $-2.55$ cm w.e. d$^{-1}$ (red-shaded area).

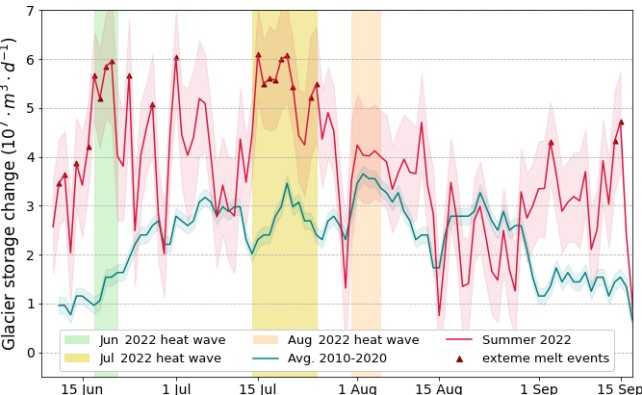

**Figure 9.** Daily storage change in all ca. 1400 Swiss glaciers. The storage change during summer 2022 (red line) is compared to the average storage change in the past decade (blue line). The red- and blue-shaded areas show the uncertainties, which are calculated as described in Sect. 5.2. Storage change rates occurring during extreme melt events, defined as days with a mass balance anomaly exceeding $-2.55$ cm w.e. d$^{-1}$, are shown by red triangles. The three heat waves that occurred in June, July, and August 2022 are shown in green, yellow, and orange, respectively.

mass balance anomalies across stations and compute their correlation for the summer seasons 2019–2021 (Fig. 10). The daily mass balance anomalies correlate well ($r = 0.52$–$0.82$; see Fig. 10a–c) for stations placed on the same glacier, i.e. FIN 1 and FIN 2 as well as RHO 1, RHO 2, and RHO 3. This is in agreement with previous studies showing that glacier mass balance is correlated well over short to intermediate distances (Lliboutry, 1974; Thibert et al., 2013; Vincent et al., 2017, 2018). The correlation generally decreases between stations that are located on different glaciers. This is expected, since stations on the same glacier are more likely to experience the same meteorological forcing. The correlations between the stations on Findelgletscher and Rhonegletscher range from 0.40–0.74, indicating that although agreement is weaker, the correlation is preserved. The correlations with the station on Glacier de la Plaine Morte, instead, are generally poor. An exception is given by the year 2020, when the correlation is similar to the one between Findelgletscher and Rhonegletscher. This lower correlation has already been noticed by Landmann et al. (2021), who suggested this to be linked to differences in local meteorological–climatic conditions (see also discussion below).

To investigate whether the uncertainties in the algorithm used to automatically extract the daily mass balances have an influence on the correlation between stations, we aggregate the mass balance anomalies to 5 d periods (Fig. 10d–f). The argument for doing so is that the uncertainties in the daily readings should be independent from each other, meaning that a reading over a longer period of time should have a smaller relative uncertainty and thus increase the correlation. Indeed, the correlations of the anomalies aggregated in this way increase for stations that are on the same glacier. Similarly, the correlations increase for combinations of stations on Findelgletscher and Rhonegletscher. The anomalies of the stations in Glacier de la Plaine Morte, instead, remain poorly correlated with other stations. The moderate increase in correlation, however, suggests that the influence of the uncertainties introduced by the automated algorithm is small, thus not affecting our spatial analysis further.

Finally, we computed the correlation between the mean anomaly of all stations and the anomaly at each individual station. This allows assessment of whether some stations are more representative than others of the regional-scale signal. The results from both the daily and the 5 d aggregated correlation matrices are congruent (Fig. 10): during the 3 years with data, the station on Glacier de la Plaine Morte shows a significantly lower correlation with the mean as compared to the other stations. The reason for this is likely to be the different local meteorological forcing: while Findelgletscher and Rhonegletscher are influenced by southerly weather patterns, Glacier de la Plaine Morte experiences weather that is more similar to the northern flanks of the Alps. This difference in meteorological forcing is corroborated by major differ-

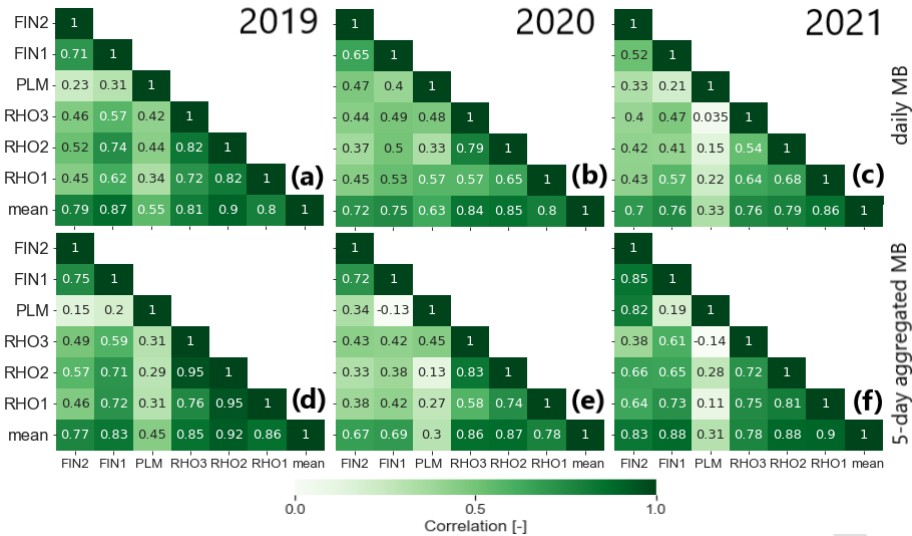

**Figure 10.** Correlation coefficient ($r$) of mass balance anomalies between the different stations. Panels **(a)**, **(b)**, **(c)** show the correlation of the daily mass balance anomalies between the six stations (indicated by their name) and the correlation between each station and the mean of all stations (labelled "mean"). Panels **(d)**, **(e)**, and **(f)** show the same but aggregated over a period of 5 d.

ences in winter snow accumulation at the stations: whereas only 1–2 m of snow are typically present in April on the tongue of Findel- and Rhonegletscher, more than 4 m are often recorded on Glacier de la Plaine Morte (Bauder et al., 2020; GLAMOS, 2022). Furthermore, the topographical situation of the station on Glacier de la Plaine Morte differs from the other ones: the station is not located at the tongue of a valley glacier, where catabatic winds influence the surface energy balance, but on a glacier plateau surrounded by mountains that might favour cold air trapping. We conclude that the relatively low number of stations and the fact that they are all located in southwestern Switzerland is limiting when aiming to infer daily mass balance variability at the regional scale, although we note that the stations are located in the region that holds the largest glacier area in the Swiss Alps (Linsbauer et al., 2021).

## 5.2 Uncertainty analysis

To estimate the overall uncertainty in the regional mass balance of summer 2022, we account for uncertainties stemming from (a) the automated reading algorithm, (b) the modelled daily mass balance for the period 2010–2020, inferred from seasonal observations at GLAMOS stakes, and (c) the average regional mass balance for the period 2010–2020, used for upscaling.

The uncertainty in the automated reading algorithm ($\sigma_{\mathrm{alg}}$) was derived by calculating the mean absolute deviation (MAD) of the automatically retrieved daily mass balances with respect to the visual readings (Fig. 11a). The average MAD for the six sites is 0.81 cm w.e. d$^{-1}$. Note that this uncertainty is biased during summer snowfall events due to the large uncertainties in the visual method in such cases.

The latter stem from (i) emergences that are very low and (ii) snow that covers some tapes of the observed stake and the ring at the base of the stake that serves as a reference. This bias is highlighted by the large deviations of the automated readings when visual readings show zero melt (Fig. 11a). In fact, the algorithm is able to capture smaller emergences than the visual readings, i.e. emergences in the order of a few pixels. Therefore, it appears that the automated method estimates higher ablation for melt rates close to 0, while the visual method fails to capture them. Capturing small emergences also allows measuring melt occurring beneath the fresh snow after summer snowfall events, i.e. when the fresh snow layer is not thick enough to entirely insulate the ice from melt.

Uncertainties in the modelled daily mass balance at the GLAMOS stakes ($\sigma_{\dot{b}}$) were derived by conducting a sensitivity analysis (Fig. 11b). To investigate the model sensitivity to various inputs, the local daily mass balance model was rerun by still ensuring a match with seasonal observations but by varying (i) the meteorological station providing temperature and precipitation forcing, (ii) the gradient to extrapolate air temperature to the site, and (iii) the ratios between the melt factor and radiation factors of the temperature-index model (see Huss and Bauder, 2009). The tested range of all variations is large and represents what we would consider a maximum offset from the reference setup, i.e. a conservative estimate for the resulting uncertainties. The deviation of the mass balance resulting from the different model runs (e.g. varying inputs i–iii) from the reference mass balance provides the uncertainties $\sigma_{\mathrm{loc},1}$, $\sigma_{\mathrm{loc},2}$, and $\sigma_{\mathrm{loc},3}$. These uncertainties are considered to be independent and are combined into an overall uncertainty in local mass balance $\sigma_{\dot{b}}$ for every

day as

$$\sigma_{\dot{b}} = \sqrt{\sigma_{\text{loc},1}^2 + \sigma_{\text{loc},2}^2 + \sigma_{\text{loc},3}^2}, \tag{8}$$

resulting in daily uncertainties in the modelled point mass balance ranging between 0.19 and 1.18 cm w.e. d$^{-1}$ (with a mean value of 0.58 cm w.e. d$^{-1}$).

Uncertainties in the average regional mass balance for the period 2010–2020 were also estimated based on a sensitivity analysis (Fig. 11c). We recomputed the 2010–2020 regional daily glacier mass balance by (i) varying the set of glaciers featuring daily glacier-wide mass balance variations, (ii) randomly superimposing uncertainties in the observation of the glacier-specific ice volume change (Fischer et al., 2015), and (iii) varying the volume-to-mass conversion factor within ±60 kg m$^{-3}$. The deviation of the mass balance resulting from the different model runs (e.g. varying inputs i–iii) from the reference mass balance provides the uncertainties $\sigma_{\text{reg},1}$, $\sigma_{\text{reg},2}$, and $\sigma_{\text{reg},3}$. Also here, we combine these uncertainties for every day as

$$\sigma_{\dot{B}_{2010\text{–}2020}} = \sqrt{\sigma_{\text{reg},1}^2 + \sigma_{\text{reg},2}^2 + \sigma_{\text{reg},3}^2}. \tag{9}$$

The uncertainty in the daily average mass balance at the scale of all Swiss glaciers $\sigma_{\dot{B}_{2010\text{–}2020}}$ is found to range between 0.06 and 0.21 cm w.e. d$^{-1}$ (with a mean value of 0.13 cm w.e. d$^{-1}$), with the highest values during the summer season.

The uncertainty in the regional mass balance of summer 2022 is then derived by combining the uncertainties in (a) $\sigma_{\text{alg}}$, (b) $\sigma_{\dot{b}}$, and (c) $\sigma_{\dot{B}_{2010\text{–}2020}}$ as

$$\sigma_{\dot{B}_{2022}} = \sqrt{\sigma_{\text{alg}}^2 + \sigma_{\dot{b}}^2 + \sigma_{\dot{B}_{2010\text{–}2020}}^2}. \tag{10}$$

The mean daily uncertainty in the regional mass balance of summer 2022 $\sigma_{\dot{B}_{2022}}$ is 0.98 cm w.e. d$^{-1}$.

Finally, we derive the uncertainty in the Switzerland-wide glacier storage change by propagating the uncertainty in the regional mass balance of summer 2022 ($\sigma_{\dot{B}_{2022}}$) and the uncertainty in the Switzerland-wide glacier area ($\sigma_A$; Linsbauer et al., 2021) with

$$\sigma_{\text{d}S} = \text{d}S \cdot \sqrt{\left(\frac{\sigma_A}{A}\right)^2 + \left(\frac{\sigma_{\dot{B}_{2022}}}{\dot{B}_{2022}}\right)^2}. \tag{11}$$

The mean daily uncertainty in the Switzerland-wide glacier storage change is found to be ±9.8 × 10$^6$ m$^3$ w.e. d$^{-1}$.

Uncertainties in the storage change cumulated over a certain period, i.e. the heat waves and the entire summer, are derived with the same procedure as described in the present section, but first we cumulate the mass balance over the period in question, and then we calculate the corresponding uncertainty assuming all mass balance uncertainties within each period to be independent.

## 5.3 Limitations and potential of the computer-vision algorithm

In this section, we discuss the limitations and possible improvements in the computer-vision algorithm that we presented. To date, the main limitation of the algorithm is that stake emergences larger than 4 cm between two images cannot be measured. This is because of the ambiguity that is introduced when the tapes move more than a complete phase shift. Indeed, for the algorithm, an emergence of (say) 4 cm looks identical to a phase shift of 8, 12, or 16 cm. This means that the method cannot be applied in cases with ablation > 4 cm between consecutive images (note that this might happen for both cases with higher overall melt rate and cases with images taken with a lower temporal resolution). There are two simple solutions for accommodating such cases: (1) placing the tapes at larger distances, although this could come at the expense of a less reliable detection since there would be fewer tapes visible on every given image, and (2) acquiring images with a time interval that is sufficiently low so as to prevent stake emergences > 4 cm between two consecutive images. A more elaborate solution would be to adapt the algorithm to use the tapes' colour coding, since this would help to resolve the ambiguity. We performed some testing by detecting the colour of the tapes, and the tests suggested potential for such an implementation (Cremona et al., 2023). An alternative for solving the problem would be to use a different feature-detection algorithm, such as scale-invariant feature transform (Lowe, 2004) or convolutional-neural-networks-based methods (Hashemi et al., 2016). Such algorithms might also be able to better deal with both rotations due to stake tilting and scaling of the tapes in the images. The reason that we preferred template matching over such potential alternatives is that the former proved to be very effective in detecting the tapes, even in challenging conditions such as fog (Fig. 12a, b) or darkness (Fig. 12c), and when the stake was partially tilted. During the development of the algorithm, we realized that the illumination conditions as well as the saturation and the colour of the tapes have an important influence on the correct functioning of the template matching procedure. In particular, the proper choice of the template was found to be key for optimal functioning, and tapes with high-contrast and saturated colours were chosen (Borner and Cremona, 2020). By contrast, tapes with low contrast and saturation (e.g. white and grey) are difficult to detect, leading to the recommendation to always use saturated, high-contrast colours (e.g. red, blue, yellow, and green). The colour choice thus represents a trade-off between (i) having many different colours (some of which with possibly low contrast and saturation) to resolve the ambiguity of emergences exceeding the tape-to-tape distance and (ii) having only a few high-contrast, saturated colours to maximize the detectability of the tapes on the individual images.

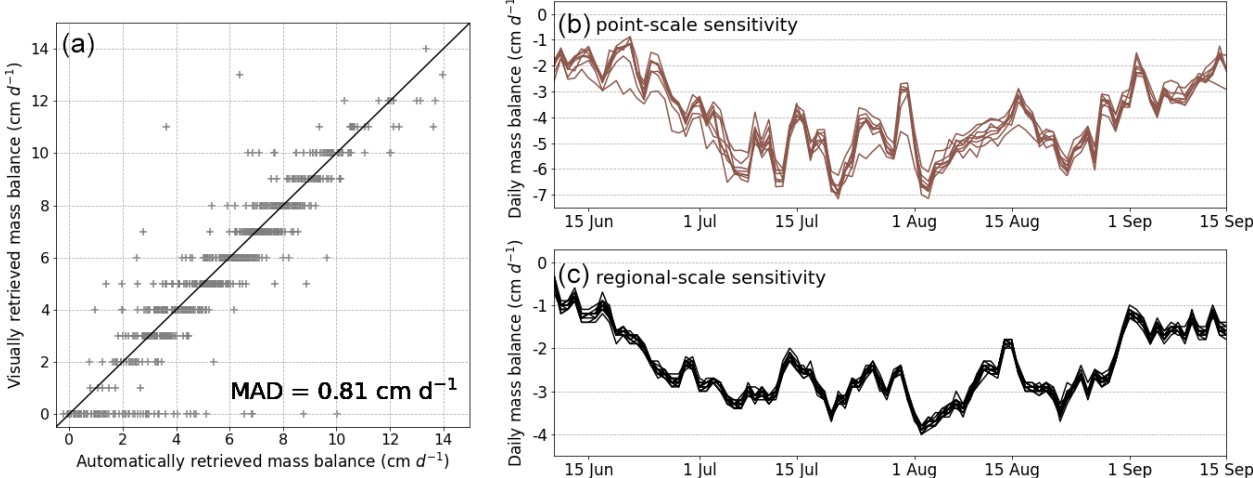

**Figure 11.** Uncertainty assessment. **(a)** Evaluation of the automated readings against the visual readings. The resolution of the visual readings is 1 cm w.e. d$^{-1}$, thus explaining the discrete values on the $y$ axis. The mean absolute deviation is 0.81 cm w.e. d$^{-1}$. **(b)** Sensitivity analysis of the modelled daily mass balance of the last decade for the GLAMOS stake close to station RHO 3. The point-scale model is run with different setups (individual brown lines) to investigate the sensitivity to various inputs. **(c)** Sensitivity analysis of the average regional mass balance for the period 2010–2020. Similar to **(b)**, the regional-scale model is run with different setups to investigate its sensitivity.

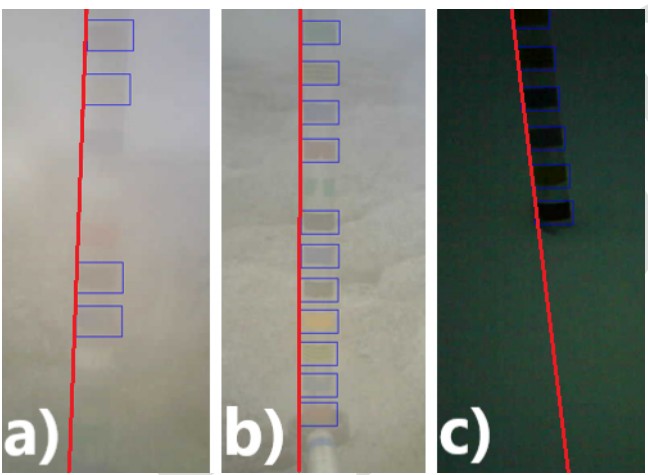

**Figure 12.** Template matching in challenging conditions: the algorithm proved to perform well and to detected many tapes even with fog **(a, b)** and darkness **(c)**.

## 5.4 Significance of heat waves

Because glacier melt that occurs during heat waves can be important for mitigating drought conditions (Van Tiel et al., 2021; Pelto et al., 2022; Ultee et al., 2022), the individual heat waves are compared to each other. The first heat wave occurred in early summer (17–21 June 2022), and glacier melt is categorized as extreme in 4 out of 5 d (Fig. 9). The glacier storage change during this heat wave accounts for $0.30 \pm 0.05$ km$^3$ w.e. Furthermore, the average daily glacier storage change during this heat wave is $4.96 \times 10^7$ m$^3$ w.e. d$^{-1}$, which is about 3 times higher than

the average daily glacier storage change in the past decade during the equivalent period. The second heat wave occurred between 14–26 July 2022, and glacier melt is categorized as extreme for 9 out of 13 d (Fig. 9). The average daily glacier storage change ($5.33 \times 10^7$ m$^3$ w.e. d$^{-1}$) is about twice as high compared to the long-term average, and the total glacier storage change caused by this heat wave is $0.68 \pm 0.06$ km$^3$ w.e. Even though the average daily glacier storage change has a similar magnitude for the June and the July heat waves ($4.96 \times 10^7$ and $5.33 \times 10^7$ m$^3$ w.e. d$^{-1}$, respectively), the difference between the daily glacier storage change in 2022 and the decadal average is much higher for the June heat wave. This is because average melt is lower in June, thus making the June heat wave result more anomalous. Nevertheless, the July heat wave was more prolonged, thus causing larger absolute glacier mass loss. The third heat wave occurred between 31 July and 6 August, but on none of the 7 d was the melt categorized as extreme (Fig. 9). The average daily glacier storage change, i.e. $4.05 \times 10^7$ m$^3$ w.e. d$^{-1}$, is about the same as compared to the long-term average, and the amount of water produced during this heat wave is $0.28 \pm 0.04$ km$^3$.

The definition of heat waves has an impact on this assessment, and other definitions – based on other temperature thresholds or on climatological temperature anomalies, for example – may be used. Even though, the chosen definition is simple, the threshold of 30 °C was used by a number of studies in Switzerland (Beniston, 2004; Beniston and Diaz, 2004) and Europe (Kovats et al., 2004; Hutter et al., 2007; Xu et al., 2016). In addition, we adopted the definition of Kyselỳ (2002) because the minimum length of 3 d on which the temperature must be above 30 °C does not exclude short

heat waves. For example, Kovats et al. (2004) used the same temperature threshold of 30 °C but a minimum number of 6 d on which the temperature must be above 30 °C, which would exclude the June heat wave that lasted 5 d.

The comparison of the three heat waves indicates that the July heat wave was the most severe. This is because of its long duration, causing about the same amount of glacier mass loss as the June and August heat waves combined. In terms of magnitude, instead, the June and July heat waves are similar, with the June heat wave being the most intense when compared to the long-term average melt. The comparison also proves the significance of melt occurring during heat waves for determining overall seasonal mass loss, as extreme heat events are expected to increase in the future (Fischer and Schär, 2010; Fischer et al., 2021). Furthermore, the large amounts of water that are produced during such short periods evidences the importance of monitoring short-term glacier mass balance variations. We suggest that this can be particularly important for optimizing the use of glacier runoff for hydropower production or for an adequate management of water supplies during heat and drought events (Zappa and Kan, 2007; Terrier et al., 2011; Anghileri et al., 2018; Landmann et al., 2021).

## 6   Conclusions

In this study, we developed a novel method for the automated derivation of daily glacier mass balance at the point scale. The method is based on computer-vision techniques, which are used to read colour-taped ablation stakes on time series of close-range camera images. A daily mean absolute deviation of $0.81\,\mathrm{cm\,w.e.\,d^{-1}}$ was found for the automated readings. By comparing the mass balance time series derived in this way with the average mass balance of the past decade, we detected extreme melt events in the summer seasons of 2019–2022. A focus was set on summer 2022, which was extraordinary in terms of glacier melt rates. Three severe heat waves were identified, comprising a total of 23 melt events that we classify as extreme. This is about as many events as had occurred over the 3 previous years, emphasizing the exceptionality of the year. Our approach also detected 25 d of heat waves (i.e. days for which maximum air temperatures averaged over four reference stations exceeded 30 °C) during summer 2022 and demonstrated a high correlation between heat waves and extreme melt events. We assessed the glacier storage change that occurred at the scale of Switzerland and found that the 25 d of heat wave occurring during summer 2022 caused an amount of melt that, based on the average over the last decade, is equivalent to 56 % of the melt expected for an entire summer. Despite the simplicity of the approach we use to upscale our point observations to the regional mass balance, we showed that extreme melt events occurring during heat days can be determinant for the total seasonal melt.

Our results evidence the importance of real-time observations for studying short-term mass balance variations and provide a means of improving our understanding of the implications that a future climate – in which more frequent heat waves and extreme melt events are expected – will have on glacier runoff. By emphasizing the large amount of meltwater that is produced during short periods, the study confirms the role that glacier runoff has in attenuating the impacts of heat waves and drought events. We suggest that establishing systems for real-time monitoring of glacier mass balance and runoff could serve to optimize the use of meltwater for hydropower production as well as to improve the management of water resources in a future, more extreme climate.

*Code and data availability.*   The code of the automated algorithm and the code used to process the data are available online at https://doi.org/10.5281/zenodo.7405281 (Cremona et al., 2023). Data are currently available at https://doi.org/10.3929/ethz-b-000602387 (Cremona, 2023).

*Video supplement.*   An example application of the automated ice ablation reading for station FIN 1 is available as a video under the following DOI: https://doi.org/10.5446/60100 (Cremona, 2022).

*Author contributions.*   AC, MH, JML, and DF conceived the study. JB and AC developed the computer-vision algorithm, with input from JML and DF. AC processed the images with the algorithm to derive the mass balance for 2019–2022. JML, MH, and AC made the visual readings of the image time series. AC wrote the paper and produced the figures, with contributions from all co-authors.

*Competing interests.*   At least one of the (co-)authors is a member of the editorial board of *The Cryosphere*. The peer-review process was guided by an independent editor, and the authors also have no other competing interests to declare.

*Acknowledgements.*   We are grateful to all persons that helped us with the installation and maintenance of the automated stations, including but not limited to Raphael Moser, Christophe Ogier, Andreas Bauder, Jane Walden, Elias Hodel, and Leo Hösli. We thank Massimiliano Zappa, Konrad Bogner, Astrid Björnsen, and Niklaus Zimmermann for the valuable collaboration within the MaLeFiX project and the WSL programme eXtremes. Further we thank the two referees that reviewed our manuscript. Their comments and inputs were very helpful in improving the final paper.

*Financial support.* This research has been supported by the programme "eXtremes" of the Swiss Federal Institute for Forest, Snow and Landscape Research (WSL) project "Machine Learning aided ForecastIng of drought related eXtremes (MaLeFiX)". CE3

*Review statement.* This paper was edited by Ben Marzeion and reviewed by two anonymous referees.

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

**Remarks from the language copy-editor**

CE1    Please give an explanation of why this needs to be changed. We have to ask the handling editor for approval. Thanks. 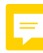

CE2    Thanks for clarifying what you mean here. "Reddish" and "blueish" cannot be used in that way, as they would be expected to correspond to a single shade which is regarded as being somewhat red or blue. Could these two instances be changed to "different shades of red" and "different shades of blue" instead? 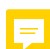

CE3    Please note minor adjustments to the punctuation. 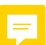

**Remarks from the typesetter**

TS1    Please provide date of last access.

TS2    Please provide date of last access.