# Peer review of "Heat wave contribution to 2022's extreme glacier melt from automated real-time ice ablation readings"

_The Cryosphere, 2022_

## Referee Comment (RC1)

General comments:

The authors showcase the potential of near real time ablation monitoring in combination with long term mass balance data and modeling to assess the impacts of heat waves on Alpine glaciers. The manuscript explains an image analysis algorithm developed by the authors to automatically process images from their camera-based ablation measurement system. In a further step, the temporally highly resolved ablation data extracted from the images are used in a more general assessment of the extremely warm 2022 ablation season, which is then compared to a decadal regional average of glacier mass balance. This is an interesting and very timely contribution that shows the immense value of high resolution, real time glacier monitoring and data assimilation for impact assessments of extreme heat events. I have a few relatively minor comments/questions that I am sure can be addressed. I look forward to seeing this work published in TC.

Mainly, I would like some more detail on the image analysis algorithm, particularly in the discussion section. The methods and results go into a fair amount of detail on the algorithm, while the discussion focuses on the 2022 season in the regional context and related uncertainty analysis. Since it is a central part of this study, I would like to see some discussion of the automated image analysis. For example, I would be interested in more context on why template matching was chosen as opposed to other feature detection algorithms that might potentially handle rotation and scaling variations better. I would also be interested in learning more about whether/how changes in illumination affect the template matching.

The authors commendably produced a "read the docs" documentation for their template matching code. I found the examples of good and bad template images instructive. In my opinion it would be worth making these easier to find by putting them in the supplement or referring the reader more directly to the code documentation. The section on "known issues" (https://rtgmc.readthedocs.io/en/latest/outlook.html) answered some additional questions I had after reading the manuscript and some of this content might be added to the discussion.

The fact that changes of more than 4 cm between images cannot be measured by the algorithm seems like a significant limitation in terms of applying the algorithm to similar use cases with slightly different measurement setups, e.g. with a lower temporal resolution and thus greater ablation between consecutive images. Could this be solved in the future by adapting the algorithm or applying some other, perhaps CNN based feature detection procedure? I understand that this is not supposed to be a computer vision paper but I think that giving a "glaciological perspective" on some common problems of feature detection and possible solutions would be a valuable addition to the discussion.

Specific comments:
Terminology:
- "stake displacement" → I stumbled over this because I associate "displacement" with stakes changing position due to ice movement. Perhaps an alternative term could be found (something like stake emergence?), otherwise I suggest clearly defining your usage of "displacement" somewhere at the beginning of the methods section.
- State somewhere what exactly you mean by "automated readings", "visual readings", "in situ readings", "manual readings". It is mostly clear from the text but having it in writing

would help the reader. "manual readings" is used interchangeably (I think?) with "visual readings". Consistent wording would be better.

Abstract
"Compared to the average course of the past decade, the 25 days of heat waves in 2022 caused a glacier mass loss that corresponds to 56% of the overall mass loss experienced on average during summers 2010-2020, demonstrating the relevance of heat waves for seasonal melt." → Consider rephrasing. I was initially unsure if "average course of the past decade" and "average during summers 2010-2020" refer to different averages. Perhaps just start the sentence with "the 25 days…" to shorten the sentence and avoid the repetition.

Introduction
L57 "In this study, we present an approach for the automated reading of the color-coded ablation stakes proposed by Landmann et al. (2021), which allows deriving daily point mass balances via the direct glaciological method" → consider rephrasing to more clearly distinguish between what Landmann did (developed camera/stake system) and what you did (automation). This sentence sounds like you "present an approach for something proposed by Landmann".

Methods
L105 "Here, we assume that the line intersecting the highest number of matches corresponds to the stake" → when you later mention projecting onto the stake axis, do you mean you are projecting onto this intersecting line, or do you detect the stake axis in another, additional step? If so, how? If not, clarify that "stake axis" is the same as this line.

Fig3, 4 and related text: How do you deal with rotation of the image in the template? Is the rotation not large enough to matter for small time steps, or do you somehow correct for this? A reference to the video supplement somewhere in this section would also be helpful.

Fig 4: if possible: add arrows or similar to the 1 count occurrences in panel d to show which matches in panel c these represent. ie, somehow mark "erroneous" distances between matches in panel c and link them with panel d to make it easier for the reader to understand the concept.

L130 "Here, dref,px is the **most frequent distance between** all possible tape combinations in an image, and dref,cm = 4 cm is the reference distance between two tapes. In the example of Figure 4 a, **the average distance between the tapes** corresponds to 42.7 px, resulting in a conversion factor of c =0.094 cm px−1 . … The impact of such differences is reduced by taking the average pixel distance"
→ is it the most frequent distance or the average distance? Unclear.

L140 "During summer snowfall, the algorithm is not able to capture accumulation, and the stake displacement, equal to zero, is assumed as the ablation."
→ Consider adding a note on why it does not work. Camera does not move and/or gets covered?

Figure 5: I find the inset arrangement of the panels counterintuitive. Perhaps this figure would work better if the panels are next to each other?

L209 "We attribute modelled daily mass balance of the closest series to every glacier of the most recent Swiss glacier inventory (Linsbauer et al., 2021) and match the daily time series to the glacier-specific annual mass balance (see above) by equally attributing the misfit to all days of the summer season (June-August)."
→ I struggle to understand this sentence. Does "closest" refer to spatial proximity of the 20 glaciers with GLAMOS data to all other glaciers? Does "see above" refer to the bias correction in the previous paragraphs? Is "misfit" the same as the bias explained earlier? If so, consider rephrasing to use the same terminology. If not, explain or otherwise clarify/rephrase.

Results
L250 and following paragraph → this explanation of how heat waves are defined might be moved to the methods section and could use some more detailed explanation. You cite a study by Hutter et al, who in turn cite Kysely (2002) for their definition of a heat wave. It would seem appropriate to cite Kysely here. Note that Kysely's (and Hutter's) usage is a little more complicated than "consecutive days above 30°C" and allows for hot periods separated by short, minor drops in temperature to be classified as one heat wave. It sounds like you did not use this part of their definition? They apply their heat wave definition to single locations and discuss heat waves in terms of their impact on excess mortality, choosing thresholds related to human well being rather than purely to climatological extremes. Your stations have a noticeable amount of altitudinal range and are situated on different sides of the Alps. Why did you choose these particular stations and a very broad, regional averaging process, rather than a more local assessment of temperatures? Given the abundance of weather stations in Switzerland, I assume meteorological data from locations much closer to your study sites would be available. Why not use stations at higher elevations that might better reflect the meteorological conditions at your study sites? You go on to state that extreme melt occurred outside of heat wave periods as per your definition - doesn't this indicate that some other definition of "heat wave" (e.g. using climatological anomalies) might be more appropriate for this application? I have no major objections to your heat wave definition in principle but I would like to see more explanation on why you decided to do it this way, particularly since "heat wave" is in the title of the paper and the concept is central to many of the arguments you present.

*Kysely J (2002) Temporal fluctuations in heat waves at Prague-Klementinum, the Czech Republic, from 1901– 1997, and their relationships to atmospheric circulation.*
*Int J Climatol 22: 33–50*

L266 The temperatures recorded during June and September 2022 were significantly higher than in the past decade and were thus able to cause extreme melt, but were not high enough to be also categorized as heat waves.

→ Rephrase to better distinguish when the June temps were a heat wave and when they were not, e.g. by giving dates for the heat wave. This is shown in fig 9 but the above sentence is hard to understand.

Fig 9 : what is the shading around the red and blue lines? uncertainty? Please add to the caption.

Discussion:
As mentioned above, I think a brief subsection discussing issues related to the template matching algorithm and how the algorithm might be adapted or improved would be appropriate. Also consider adding a few comments on how different definitions of heat waves might affect the overall calculations of storage change.

Typos
BE / AE spelling not always consistent (e.g. modeling / modelling)
L24 (Patro et al., 2018; Schaefli et al., 2019)) → remove extra parentheses

L26
"Despite glacier mass balance has been studied extensively with remote sensing (Bamber and Rivera, 2007), in-situ observations (Zemp et al., 2009), and modelling approaches (Hock, 2005; Hock et al., 2019), daily-scale mass balance variations remain mostly unexplored." → "despite" should be followed by a noun or gerund, consider replacing "despite" with "although" or rephrasing.

L54 Because these novel methods **allows** measuring → allow

2 Study **site** and field data → study sites? (plural)

L101 The choice of a low correlation threshold **has also** some drawbacks. → also has (position adverb between subject and verb)

L151 By comparing the **algorithms** outcomes with the visual image readings, the daily errors of the automated approach can be computed. By comparing the **algorithms'** outcomes with the in-situ observations, the seasonal mean-absolute deviation (MAD) is derived. → algorithm's

L231 Of note is the fact that the daily deviations are mostly negative, which might suggest that the automated algorithm **overestimate** melt when compared to the visual readings. → overestimates

---

## Author Comment (AC1)

**Answer to Reviewer 1**

In the following, we provide a point-by-point Author Response (AR) to any of the Reviewer Comments (RC) obtained for the manuscript that was under discussion. When presenting suggestions for how the manuscript text could be revised (*italic text* in quotation marks), the line numbers refer to the revised manuscript.

**RC1**: Mainly, I would like some more detail on the image analysis algorithm, particularly in the discussion section. The methods and results go into a fair amount of detail on the algorithm, while the discussion focuses on the 2022 season in the regional context and related uncertainty analysis. Since it is a central part of this study, I would like to see some discussion of the automated image analysis. For example, I would be interested in more context on why template matching was chosen as opposed to other feature detection algorithms that might potentially handle rotation and scaling variations better. I would also be interested in learning more about whether/how changes in illumination affect the template matching. The authors commendably produced a "read the docs" documentation for their template matching code. I found the examples of good and bad template images instructive. In my opinion it would be worth making these easier to find by putting them in the supplement or referring the reader more directly to the code documentation. The section on "known issues" (https://rtgmc.readthedocs.io/en/latest/outlook.html) answered some additional questions I had after reading the manuscript and some of this content might be added to the discussion. The fact that changes of more than 4 cm between images cannot be measured by the algorithm seems like a significant limitation in terms of applying the algorithm to similar use cases with slightly different measurement setups, e.g. with a lower temporal resolution and thus greater ablation between consecutive images. Could this be solved in the future by adapting the algorithm or applying some other, perhaps CNN based feature detection procedure? I understand that this is not supposed to be a computer vision paper but I think that giving a "glaciological perspective" on some common problems of feature detection and possible solutions would be a valuable addition to the discussion.

**AR1**: We thank the Reviewer for the thorough review of our manuscript and for the time invested in checking our submission. It is rare to see reviewers checking the documentation provided to a given code, and we truly appreciate the time investment.

For what the specific comment is concerned, we agree with the Reviewer that more detailed discussion can be provided for the automated image analysis. In the revised manuscript, we intend to do so with the following section in the discussion:

Lines XXX "***5.3 Limitations and potential of the computer vision algorithm***

*In this section, we discuss the limitations and possible improvements of the computer vision algorithm that we presented. To date, the main limitation of the algorithm is that stake emergences larger than 4 cm between two images cannot be measured. This is because of the ambiguity that is introduced when the tapes move more than a complete phase shift. Indeed, for the algorithm, an emergence of (say) 4cm looks identical to a phase shift of 8, 12, or 16 cm. This means that the method cannot be applied in cases with ablation > 4cm between consecutive images (note that this might happen for both cases with higher overall melt rate and cases with images taken with lower temporal resolution). There are two simple solutions for accommodating such cases: 1) place the tapes at larger distances, although this could come at the expense of a less reliable detection, since there would be less tapes visible on every given image, and 2) acquire images with a time interval that is sufficiently low as to*

*prevent stake emergences > 4 cm between two consecutive images. A more elaborate solution would be to adapt the algorithm to use the tapes' color coding, since this would help to resolve the ambiguity. We performed some testing by detecting the color of the tapes, and the tests suggested potential for such an implementation (Cremona et al. 2022). An alternative for solving the problem would be to use a different feature-detection algorithm, such as Scale-Invariant Feature Transform (Lowe 2004) or Convolutional Neural Networks based methods (Hashemi et al. 2016). Such algorithms might also be able to better deal with both rotations due to stake tilting and scaling of the tapes in the images. The reason that we preferred template matching over such potential alternatives is that the former proved to be very effective in detecting the tapes, even in challenging conditions such as fog (Fig, Aa, b) or darkness (Fig Ac), and when the stake was partially tilted.*

*During the development of the algorithm, we realized that the illumination conditions, as well as the saturation and the color of the tapes, have an important influence on the correct functioning of the template matching procedure. In particular, the proper choice of the template was found to be key for optimal functioning, and tapes with high-contrast and saturated colors were chosen (Borner and Cremona 2020). On the contrary, tapes with low contrast and saturation (e.g. white and grey) are difficult to detect, leading to the recommendation of always using saturated, high-contrast colors (e.g. red, blue, yellow, and green). The color choice thus represents a trade-off between (i) having many different colors (among which some with possibly low contrast and saturation) to resolve the ambiguity of displacements exceeding the tape-to-tape distance, and (ii) having only a few high-contrast, saturated colors to maximize the detectability of the tapes on the individual images."*

[Figure]

*Figure A: Template matching in challenging conditions: the algorithm proved to perform well and to detected many tapes even with fog (a, and b) and darkness (c).*

**RC2**: "stake displacement" → I stumbled over this because I associate "displacement" with stakes changing position due to ice movement. Perhaps an alternative term could be found (something like stake emergence?), otherwise I suggest clearly defining your usage of "displacement" somewhere at the beginning of the methods section.

**AR2**: This is a valid point also mentioned by Reviewer #2. To avoid confusion, we decided to replace "stake displacement" with "stake emergence", as suggested above.

**RC3**: State somewhere what exactly you mean by "automated readings", "visual readings", "in situ readings", "manual readings". It is mostly clear from the text but having it in writing would help the reader. "manual readings" is used interchangeably (I think?) with "visual readings". Consistent wording would be better.

**AR3**: We apologize for the confusion caused by the inconsistent and partially undefined wording. We suggest rephrasing L148-151 as follows (note that we no longer use the term "manual readings", which indeed we used interchangeably with "visual readings"):

LXXX: "*For validation, the automated readings ( i.e. the outcomes of the algorithm), were compared to (i) visual readings of the same image time series (performed following Landmann et al. 2021), and (ii) in-situ stake readings obtained by following the glaciological method. In-situ stake readings were conducted two to five times per season in conjunction with the installation and maintenance of the stations, as well as with other field campaigns.*"

**RC4**: Abstract "Compared to the average course of the past decade, the 25 days of heat waves in 2022 caused a glacier mass loss that corresponds to 56% of the overall mass loss experienced on average during summers 2010-2020, demonstrating the relevance of heat waves for seasonal melt." → Consider rephrasing. I was initially unsure if "average course of the past decade" and "average during summers 2010-2020" refer to different averages. Perhaps just start the sentence with "the 25 days…" to shorten the sentence and avoid the repetition.

**AR4**: Rephrased as suggested: LXXX: "*The same 25 days of heat waves caused a glacier mass loss that corresponds to 56% of the average mass loss experienced over the entire melt season during the summers 2010-2020, demonstrating the relevance of heat waves for seasonal melt.*"

**RC5**: Introduction L57 "In this study, we present an approach for the automated reading of the color-coded ablation stakes proposed by Landmann et al. (2021), which allows deriving daily point mass balances via the direct glaciological method" → consider rephrasing to more clearly distinguish between what Landmann did (developed camera/stake system) and what you did (automation). This sentence sounds like you "present an approach for something proposed by Landmann".

**AR5**: Rephrased as follows:

LXXX: "*In this study, we present an approach for the automated reading of the color-coded ablation stakes that were presented by Landmann et al. (2021). Our approach allows deriving daily point mass balances automatically via the direct glaciological method from a time series of close-range images depicting a given ablation stake.*"

**RC6**: Methods L105 "Here, we assume that the line intersecting the highest number of matches corresponds to the stake" → when you later mention projecting onto the stake axis, do you mean you are projecting onto this intersecting line, or do you detect the stake axis in another, additional step? If so, how? If not, clarify that "stake axis" is the same as this line.

**AR6**: To clarify this point, we rephrased as follows:

LXXX: "*Exploiting this property, the difference in position for all possible combinations is calculated, and projected onto the stake axis (red line in Fig. 2) with the following equation:*"

**RC7**: Fig3, 4 and related text: How do you deal with rotation of the image in the template? Is the rotation not large enough to matter for small time steps, or do you somehow correct for this? A reference to the video supplement somewhere in this section would also be helpful.

**AR7**: The stake is quite stable throughout the season, i.e. the tilt to the side is typically low, and rotation is minimal. Therefore, the correlation matching works well without any correction. We agree that if the tilt of the stake is large, the algorithm may struggle, but at this point the measurement of the mass balance itself (even when reading visually) is compromised. So far, we didn't encounter any problem in this respect. In the newly added Section 5.3 "Limitations and potential of the computer vision algorithm" (cf. AR1), we discuss in more detail how the algorithm deals with rotation (see our AR1 at that point).

For the last part of the comment, we now added the reference to the video supplement: LXXX: "*An example of application showing the automated ablation reading on Findelgletscher can be found in the digital supplementary of this article*".

**RC8**: Fig 4: if possible: add arrows or similar to the 1 count occurrences in panel d to show which matches in panel c these represent. ie, somehow mark "erroneous" distances between matches in panel c and link them with panel d to make it easier for the reader to understand the concept.

**AR8:** We added the arrows (Fig. B) and adjusted the figure caption accordingly. The new figure might be a bit busier but makes the procedure clearer. Therefore, we agree on substituting the original one with Figure B.

[Figure]

*Figure B: Stake displacement calculation. (a) Filtered matches in image i-1 in blue. (b) Filtered matches in image i in yellow. (c) The two sets of matches are superimposed. (d) Occurrence of displacements dk resulting from the difference of the positions between all possible combinations of tapes. As an example, the top match in image i-1 (panel a) is compared with all the tapes in image i (panel b). The red lines show combinations of matches that provide displacements dk that are different than the stake emergence, whereas the green line provides the correct displacement. The displacement of each combination is reported in panel d (green and red horizontal lines). The procedure is repeated for every match in image i, and the most frequent displacement (0.668 pixels in the example) is taken as stake emergence.*

**RC9**: L130 "Here, dref,px is the most frequent distance between all possible tape combinations in an image, and dref,cm = 4 cm is the reference distance between two tapes. In the example of Figure 4 a, the average distance between the tapes corresponds to 42.7 px, resulting in a conversion factor of c =0.094 cm px−1 . … The impact of such differences is reduced by taking the average pixel distance" → is it the most frequent distance or the average distance? Unclear.

**AR9**: This was our mistake, as it is the average distance. We replaced "most frequent" with "average distance", and the sentence now reads:

LXXX: "*Here, dref,px is **the average distance** between the tape in an image, and dref,cm = 4 cm is the reference distance between two tapes. In the example of Figure 4 a, the average distance between the tapes corresponds to 42.7 px, resulting in a conversion factor of c =0.094 cm px−1 . … The impact of such differences is reduced by taking the average pixel distance*"

**RC10**: L140 "During summer snowfall, the algorithm is not able to capture accumulation, and the stake displacement, equal to zero, is assumed as the ablation." → Consider adding a note on why it does not work. Camera does not move and/or gets covered?

**AR10**: For sake of clarity and according to the Reviewer's suggestion, we rephrased as follows:

LXXX: "*During summer snowfall, the algorithm is not able to capture accumulation because the snow which accumulates on the surface covers the camera but does not cause any relative movement between the camera and the stake. The stake displacement is thus equal to zero, which is also our ablation reading.*"

**RC11**: Figure 5: I find the inset arrangement of the panels counterintuitive. Perhaps this figure would work better if the panels are next to each other?

**AR11**: We agree that the inset is not the most intuitive arrangement. Originally the two panels were placed next to each other, but the image occupied a lot of space, most of this space (the one in which panel b is located now) being unused. To avoid such a space-consuming arrangement, prefer to keep the image as it is now.

**RC12**: L209 "We attribute modelled daily mass balance of the closest series to every glacier of the most recent Swiss glacier inventory (Linsbauer et al., 2021) and match the daily time series to the glacier-specific annual mass balance (see above) by equally attributing the misfit to all days of the summer season (June-August)." → I struggle to understand this sentence. Does "closest" refer to spatial proximity of the 20 glaciers with GLAMOS data to all other glaciers? Does "see above" refer to the bias correction in the previous paragraphs? Is "misfit" the same as the bias explained earlier? If so, consider rephrasing to use the same terminology. If not, explain or otherwise clarify/rephrase.

**AR12**: Thanks for pointing out this imprecise formulation. We have now rewritten the corresponding paragraph to make sure that the information is revealed in a clearer way.

LXXX: "*We attribute modelled daily mass balance of the closest of the 20 glaciers with dircect mass balance measurements to every glacier of the most recent Swiss glacier inventory (Linsbauer et al., 2021). Subsequently, we match the daily time series to the glacier-specific annual mass balance as originating from the procedure described above (Grab et al., 2021). This is achieved by equally attributing the misfit between the cumulative daily series and the glacier-specific annual mass balance to all days of the summer season (June-August).*"

**RC13**: Results L250 and following paragraph → this explanation of how heat waves are defined might be moved to the methods section and could use some more detailed explanation. You cite a study by Hutter et al, who in turn cite Kysely (2002) for their definition of a heat wave. It would seem appropriate to cite Kysely here. Note that Kysely's (and Hutter's) usage is a little more complicated than "consecutive days above 30°C" and allows for hot periods separated by short, minor drops in temperature to be classified as one heat wave. It sounds like you did not use this part of their definition? They apply their heat wave definition to single locations and discuss heat waves in terms of their impact on excess mortality, choosing thresholds related to human well being rather than purely to climatological extremes. Your stations have a noticeable amount of altitudinal range and are situated on different sides of the Alps. Why did you choose these particular stations and a very broad,

regional averaging process, rather than a more local assessment of temperatures? Given the abundance of weather stations in Switzerland, I assume meteorological data from locations much closer to your study sites would be available. Why not use stations at higher elevations that might better reflect the meteorological conditions at your study sites? You go on to state that extreme melt occurred outside of heat wave periods as per your definition - doesn't this indicate that some other definition of "heat wave" (e.g. using climatological anomalies) might be more appropriate for this application? I have no major objections to your heat wave definition in principle but I would like to see more explanation on why you decided to do it this way, particularly since "heat wave" is in the title of the paper and the concept is central to many of the arguments you present.

**AR13**: We thank the reviewer for this comment. Our reasoning for having the definition of the heat waves at former L. 250 was that it was only required at that stage. In hindsight, we agree that the definition might be better placed in the methods section, which we now did by adding the subsection "3.5 Definition of heat waves" (see below). Similarly, we now directly cite Kysely (2002), instead of Hutter (2007).

LXXX: ***3.5 Definition of heat waves***

*To investigate the effect of heat waves on glacier melt at the scale of the Swiss Alps, we follow the definition of Kysely (2002). According to Kysely (2002), heat waves are defined as consecutive periods of at least three days during which the average daily maximum temperature is higher than 30 °C and the daily maximum temperature on individual days does not fall below 25°C. This second part of the definition allows for hot periods separated by short, minor drops in temperature to be classified as one heat wave. Note that at the meteorological stations that we considered, the maximum daily temperature during the heat waves never lowered below 30°C, therefore the second part of the definition was not used. The definition of Kysely (2002) considered the temperature at a single station. However, because we are interested to have a representative result at the Swiss-wide scale, we considered the average temperature measured at different sites for defining heat waves. The four sites were chosen to be the meteorological stations in Lugano (273 m a.s.l.), Chur (556 m a.s.l.), Zurich (426 m a.s.l.), and Sion (482 m a.s.l.), because they are scattered over Switzerland and cover a reasonable altitudinal range (in which we can expect temperatures above 30°C). Hence, we assume the average temperature across the four stations to be representative for the regional scale."*

Regarding the second part of the comment, related to Kysely's definition: we note that the temperature during the heat waves as we defined them does not drop below 30 °C; this means that the part of Kysely's definition that allow for short drops in temperature is actually not relevant. We now state this explicitly in the new subsection 3.5 where we define the heat waves.

The above said, we acknowledge that a different definition of heat waves (e.g. one that would use a different temperature threshold, or one that considers climatological anomalies as pointed out by the Reviewer, for example) would affect our results. We therefore suggest adding the following note:

LXXX: "*The definition of heat waves has an impact on this assessment, and other definitions – based on other temperature thresholds or on climatological temperature anomalies, for example – may be used. Even though, the chosen definition is simple, the threshold of 30°C was used by a number of studies in Switzerland (Beniston 2004, Beniston and Diaz 2004) and Europe (Kovats 2004, Hutter 2007, Xu 2016). In addition, we adopted the definition of Kysely (2002) because the minimum length of three*

*days during which the temperature must be above 30°C does not exclude short heat waves. For example, Kovats (2004) used the same temperature threshold of 30°C but a minimum number of six days during which the temperature must be above 30°C, which would exclude the June heat wave that lasted five days."*

**RC14**: L266 The temperatures recorded during June and September 2022 were significantly higher than in the past decade and were thus able to cause extreme melt, but were not high enough to be also categorized as heat waves. → Rephrase to better distinguish when the June temps were a heat wave and when they were not, e.g. by giving dates for the heat wave. This is shown in fig 9 but the above sentence is hard to understand.

**AR14**: We apologize for the unclear sentence. We suggest rephrasing as follows:

LXXX: *"The temperatures recorded during June and September 2022 were significantly higher than in June-September of the past decade and thus caused extreme melt for several days. However, before the first heat wave (which started on 17 June 2022), and during September 2022, the temperatures were never high enough as to be categorized as heat waves."*

**RC15**: Fig 9 : what is the shading around the red and blue lines? uncertainty? Please add to the caption.

**AR15**: Exactly, it is the uncertainty. We suggest rephrasing the caption as follows:

*"Daily storage change of all ca. 1400 Swiss glaciers. The storage change during summer 2022 (red line) is compared to the average storage change of the past decade (blue line). **The red- and blue-shaded areas show the uncertainties, which are calculated as described in Sect. 5.2.** […]"*

**RC16**: Discussion: As mentioned above, I think a brief subsection discussing issues related to the template matching algorithm and how the algorithm might be adapted or improved would be appropriate. Also consider adding a few comments on how different definitions of heat waves might affect the overall calculations of storage change.

**AR16**: We added a subsection discussing the computer-vision algorithm as per earlier comment (see AR1). Following the Reviewer's suggestions, we also added a note on how different definitions affects the results (see AR13).

**RC17**:

Typos

BE / AE spelling not always consistent (e.g. modeling / modelling)

L24 (Patro et al., 2018; Schaefli et al., 2019)) → remove extra parentheses

L26 "Despite glacier mass balance has been studied extensively with remote sensing (Bamber and Rivera, 2007), in-situ observations (Zemp et al., 2009), and modelling approaches (Hock, 2005; Hock et al., 2019), daily-scale mass balance variations remain mostly unexplored." → "despite" should be followed by a noun or gerund, consider replacing "despite" with "although" or rephrasing.

L54 Because these novel methods allows measuring → allow

2 Study site and field data → study sites? (plural)

L101 The choice of a low correlation threshold has also some drawbacks. → also has (position adverb between subject and verb)

L151 By comparing the algorithms outcomes with the visual image readings, the daily errors of the automated approach can be computed. By comparing the algorithms' outcomes with the in-situ observations, the seasonal mean-absolute deviation (MAD) is derived. → algorithm's

L231 Of note is the fact that the daily deviations are mostly negative, which might suggest that the automated algorithm overestimate melt when compared to the visual readings. → overestimates

**AR17**: All the typos were corrected following the suggestions made by the Reviewer. Similarly, we now consistently use British English spelling.

**New References**

Hashemi, N. S., Aghdam, R. B., Ghiasi, A. S., & Fatemi, P. (2016). Template Matching Advances and Applications in Image Analysis. ArXiv. https://doi.org/10.48550/arXiv.1610.07231

D. G. Lowe, "Distinctive image features from scale-invariant keypoints", International Journal of Computer Vision, vol. 60, no. 2, pp. 91-110, 2004.

Cremona, A., Huss, M., Landmann, J. M., Borner, J., and Farinotti, D.: Heat wave contribution to 2022's extreme glacier melt from automated real-time ice ablation readings [code], https://doi.org/10.5281/zenodo.7405281, 2022.

Borner, J., Cremona, A.: Determine real-time glacier mass changes from camera images [code documentation], https://rtgmc.readthedocs.io/en/latest/index.html, 2020.

Kysely, J: Temporal fluctuations in heat waves at Prague-Klementinum, the Czech Republic, from 1901– 1997, and their relationships to atmospheric circulation. Int J Climatol 22: 33–50, 2002.

Beniston, M.,Diaz, H. F.: The 2003 heat wave as an example of summers in a greenhouse climate? Observations and climate model simulations for Basel, Switzerland. Global and Planetary Change, Volume 44, Issues 1–4, Pages 73-81, https://doi.org/10.1016/j.gloplacha.2004.06.006, 2004.

Xu, Z., FitzGerald, J., Guo, Y., Jalaludin, B., Tong, S.: Impact of heatwave on mortality under different heatwave definitions: A systematic review and meta-analysis. Environment International, Volumes 89–90, Pages 193-203, https://doi.org/10.1016/j.envint.2016.02.007, 2016.

Beniston, M.: The 2003 heat wave in Europe: A shape of things to come? An analysis based on Swiss climatological data and model simulations. Geophysical research letters 31 (2), https://doi.org/10.1029/2003GL018857, 2004.

Kovats, R. S., Hajat, S., Wilkinson, P.: Contrasting patterns of mortality and hospital admissions during hot weather and heat waves in Greater London, UK. Occupational and environmental medicine 61.11, https://doi.org/10.1136/oem.2003.012047, (2004).

---

## Author Comment (AC2)

**Answer to Reviewer 2**

In the following, we provide a point-by-point Author Response (AR) to any of the Reviewer Comments (RC) obtained for the manuscript that was under discussion. When presenting suggestions for how the manuscript text could be revised (*italic text* in quotation marks), the line numbers refer to the revised manuscript.

**RC1**: Terminology: For the process that the stake melts out of the ice due to ablation and hence increasing its length of the "free end", the author use the term (vertical) displacement (L52 and others). This causes some confusion as displacement is usually connected to ice flow, which is not intended to be detected in the presented method and apart from tilting the ablation stake has no direct influence here. I suggest using the term surface elevation change (or something equivalent) instead.

**AR1**: This is a good point, and we realize that our wording could cause confusion. Also taking into account the comment of Reviewer #1, we suggest to replace "stake displacement" with "stake emergence". We believe that "stake emergence" is better suited than "surface elevation change" since the latter wording is often used in studies of geodetic mass balance to indicate the surface elevation change at a given position, i.e. at a set of fixed coordinates. In our case, instead, we are measuring the surface mass balance of a point that is advected downstream via the glacier's ice flow. We note that our stake emergence is not to be confused with the concept of "emergence velocity", which is the difference between the local elevation change and mass balance. To avoid this latter confusion, we added the following sentence to the manuscript:

L XXX: *"To automatically read the stake, a computer-vision algorithm is used to derive the stake emergence between two images. Note that the stake emergence is the vertical movement of the stake out of the ice and is not to be confused with the concept of "emergence velocity", i.e. the* difference between the local elevation change and mass balance*."*

**RC2**: Preselection of images: In the method section the authors describe that images are taken every 20 minutes. I wonder if the method analyses all images or if a preselection using which criteria was applied.

**AR2**: Correct, the images are acquired every 20 minutes when there is enough daylight while during night, the camera does not acquire images. For the method's application, we do not perform any image preselection, i.e. we processes all of them. To better clarify this, we suggest rephrasing L80 as follows:

Line XXX: *"To automatically read the stake, all images acquired during the season are processed with a computer-vision algorithm that derives the stake emergence between pairs of subsequent images."*

**RC3**: Threshold for extreme melt event (L185-196): In Tab. 1 the authors show that there is an altitude difference of almost 800 m between the stations used for deriving daily ablation values. This difference might largely explain the spread in the mass balance anomaly in Fig. 8. However, the altitudinal distribution of the stations is skewed to lower altitudes and thus taking the mean of the stations for defining the threshold should be reconsidered. I think the median is more significant than

the mean, although the number of stations is low. Speaking of which, the number of observations presumably might change over the ablation period, as higher stations experience a longer snow cover. There should be a note how the number of observations affects the interpretation of the mass balance anomaly.

**AR3**: We disagree with the first part of this comment, or we do not exactly understand it. In Figure 8, we cannot identify a trend by which the spread in the mass balance anomaly would be larger for lower elevations. To better show this, Figure A (here below) shows the standard deviation of the anomaly of every station against its elevation. If we understand the Reviewer's comment correctly, this figure should show a trend towards higher standard deviation for lower elevations, which is apparently not the case. To us, this observation makes sense because the anomaly we compute (see Sect. 3.3) is related to the average mass balance at that given station. This means that any difference in average mass balance (which, we agree, is directly determined by the station's elevation) does not play a role anymore.

[Figure]

*Figure A: Standard deviation of the mass balance anomaly against elevation.*

For the second part of the comment, related to the number of stations, we reworded as follows to note that the number of observations influences the interpretation of the mass balance anomaly:

LXXX: "*The 85% quantile corresponds to −2.55 cm w.e d−1, and the event is classified as extreme when the mean anomaly across the stations exceeds this threshold. Because the stations have different operation periods (Tab. 2), the mean anomaly may be computed from a different number of stations and consequently be affected.*"

**RC4**: L32-34: I do not agree with these two sentences. A number of studies assess reasonable short term mass balance variations from geodetic measurements (e.g., Klug et al., 2018; Zeller et al., 2022; Beraud et al., 2022; Vincent et al., 2021). Consider rephrasing or omitting.

**AR4**: Thanks for the specification. We suggest rephrasing as follows:

LXXX: "*Interpreting geodetic mass balances at short time scales can be challenging, though, since the results are sensibly affected by the choice of the volume-to-mass conversion factor (Huss, 2013). Recent studies have nevertheless been using results from geodetic studies to gain insights on short-term mass balance variations (e.g., Klug et al., 2018; Vincent et al., 2021; Zeller et al., 2022; Beraud et al., 2022).*"

**RC5**: L77: Please rephrase and consider the width of the tape as well.

**AR5**: We suggest rephrasing as follows:

LXXX: "*The station setup consists of a camera and an aluminum stake that is marked with tapes of different colors. The tapes have a width of 2 cm and are placed 2 cm apart (Fig. 2).*"

**RC6**: Eq. 1: Please explain how you determine the stake inclination. Is it measured during the field visits or derived from the images or…?

**AR6**: The stake inclination is detected automatically by the algorithm, i.e. the angle between the red line in Figure 2 and the vertical axis). We clarified this with the following wording:

LXXX: "*Here, α is the stake inclination with respect to the vertical axis, i.e. the angle between the red line in Figure 2 and the vertical axis. The inclination is derived automatically by the algorithm […]*"

**RC7**: L257: Consider depicting these periods also in Fig. 8.

**AR7**: We thank the Reviewer for the suggestion. However, since Figure 8 shows the four different years, we think that only depicting heat waves in 2022 could be misleading. We thus prefer not to show this information in the Figure.

**RC8**: L305-307: Please better explain how winter snow accumulation impacts melt anomalies between individual stations. Winter snow accumulation might have an influence on the length of the ablation season, but how does it alter the ablation anomaly during core summer, when winter snow has melted since long?

**AR8**: The reviewer's question seems to be triggered by a misunderstanding: we agree that the winter snow accumulation has no direct effect on the summer melt rates (an indirect effect could be the albedo of the ice, which is arguably different for locations in which the snow cover is present over a longer period of time). Our point, however, was not the presence of the snow as such, but rather the different snow amounts, which indicate a difference in local meteorology. To avoid the possible misunderstanding, we reworded the sentence as follows:

L XXX: *"The reason for this is likely to be the different local meteorological forcing: while Findelgletscher and Rhonegletscher are influenced by southerly weather patterns, Glacier de la Plaine Morte experiences weather that is more similar to the Northern flanks of the Alps. This difference in meteorological forcing is corroborated by major differences in winter snow accumulation at the stations: whereas only 1-2m of snow are typically present in April on the tongue of Findel- and Rhonegletscher, more than 4m are often recorded on Glacier de la Plaine Morte […]"*

**New References**

Beraud, L., Cusicanqui, D., Rabatel, A., Brun, F., Vincent, C., and Six, D.: Glacier-wide seasonal and annual geodetic mass balances from Pléiades stereo images: application to the Glacier d'Argentière, French Alps, Journal of Glaciology, 1–13, https://doi.org/10.1017/jog.2022.79, 2022.

Klug, C., Bollmann, E., Galos, S. P., Nicholson, L., Prinz, R., Rieg, L., Sailer, R., Stötter, J., and Kaser, G.: Geodetic reanalysis of annual glaciological mass balances (2001-2011) of Hintereisferner, Austria, The Cryosphere, 12, 833–849, https://doi.org/10.5194/tc-12-833-2018, 2018.

Vincent, C., Cusicanqui, D., Jourdain, B., Laarman, O., Six, D., Gilbert, A., Walpersdorf, A., Rabatel, A., Piard, L., Gimbert, F., Gagliardini, O., Peyaud, V., Arnaud, L., Thibert, E., Brun, F., and Nanni, U.: Geodetic point surface mass balances: a new approach to determine point surface mass balances on glaciers from remote sensing measurements, The Cryosphere, 15, 1259–1276, https://doi.org/10.5194/tc-15-1259-2021, 2021.

Zeller, L., McGrath, D., Sass, L., O'Neel, S., McNeil, C., and Baker, E.: Beyond glacierwide mass balances: parsing seasonal elevation change into spatially resolved patterns of accumulation and ablation at Wolverine Glacier, Alaska, Journal of Glaciology, 1–16, https://doi.org/10.1017/jog.2022.46, 2022.